# On correlation functions in models related to the Temperley-Lieb algebra

Kohei Fukai[1], Raphael Kleinemühl[2], Balázs Pozsgay[3*] and Eric Vernier[4]

**1** The Institute for Solid State Physics, The University of Tokyo,
5-1-5 Kashiwanoha, Kashiwa, Chiba, 277-8581 Japan
**2** Fakultät für Mathematik und Naturwissenschaften,
Bergische Universität Wuppertal, Wuppertal, Germany
**3** MTA-ELTE "Momentum" Integrable Quantum Dynamics Research Group,
Department of Theoretical Physics, Eötvös Loránd University, Budapest, Hungary
**4** Laboratoire de Probabilités, Statistique et Modélisation
CNRS - Univ. Paris Cité - Sorbonne Univ. Paris, France

★ pozsgay.balazs@ttk.elte.hu

## Abstract

We deal with quantum spin chains whose Hamiltonian arises from a representation of the Temperley-Lieb algebra, and we consider the mean values of those local operators which are generated by the Temperley-Lieb algebra. We present two key conjectures which relate these mean values to existing literature about factorized correlation functions in the XXZ spin chain. The first conjecture states that the finite volume mean values of the current and generalized current operators are given by the same simple formulas as in the case of the XXZ chain. The second conjecture states that the mean values of products of Temperley-Lieb generators can be factorized: they can expressed as sums of products of current mean values, such that the coefficients in the factorization depend neither on the eigenstate in question, nor on the selected representation of the algebra. The coefficients can be extracted from existing work on factorized correlation functions in the XXZ model. The conjectures should hold for all eigenstates that are non-degenerate with respect to the local charges of the models. We consider concrete representations, where we check the conjectures: the so-called golden chain, the $Q$-state Potts model, and the trace representation. We also explain how to derive the generalized current operators from concrete expressions for the local charges.



# 1 Introduction

One dimensional quantum integrable models are special systems, which allow for exact solutions, at least for certain physical quantities and in certain physical situations [1, 2]. The Bethe Ansatz is a central method which is used to solve many such models. The strength of the Bethe Ansatz lies in diagonalizing the Hamiltonian: in finding the eigenvectors and the associated eigenvalues. However, the computation of correlation functions is a notoriously difficult problem, already in equilibrium situations.

In this work, we contribute to the computation of correlation functions in a selected class of integrable models: those quantum spin chains, which are related to the Temperley-Lieb algebra.

The Temperley-Lieb algebra was discovered in the paper [3], where it was used to compute the partition function of selected 2D statistical physical models by relating them to the XXZ Heisenberg spin chain. The key observation is that the defining relations of the algebra are strong enough to guarantee equivalences in the spectrum of different models. These models are then seen as different representations of the same algebra. The precise statement is that the eigenvalues of the Hamiltonians in some representation of the Temperley-Lieb algebra are included in the spectrum of the XXZ model (with the same volume, and open boundary conditions). However, the degeneracies of the states can differ in the various representations [4–6] (see also [7]).

As opposed to the many studies dealing with the spectrum and the representation theory, the correlation functions of these models have received far less attention. In fact, we are not aware of any work dealing systematically with this problem. Specific correlation functions in Temperley-Lieb models have been investigated in selected cases [8, 9], but we are not aware of a general treatment. Meanwhile, correlation functions of the XXZ spin chain have been investigated for multiple decades (see the book [10] and the habilitation thesis [11]). This leads to the idea of utilizing the Temperley-Lieb algebra to make connections between the XXZ chain and other representations also on the level of the correlation functions, thus yielding useful results for many models for which there are no results for correlators in the literature.

It is our goal to use the theory of factorized correlation functions of the XXZ chain (also known as the "hidden Grassmann structure") to compute correlations for other Temperley-Lieb models. In the Heisenberg chain, this theory was initiated in [12], where it was observed that certain multiple integrals for correlation functions in the XXX chain factorize: they can be expressed as sums of products of single integrals. This led to the development of a complete algebraic theory, which led to convenient factorized formulas for the mean values of short-range operators [13–20]. On a practical level, the theory states that mean values of short-range correlators can be expressed using the Taylor coefficients of a few number of functions of one and two variables. The theory consists of two parts: the algebraic part expresses the correlation functions using sums of products of these building blocks, whereas the physical part gives concrete values to the building blocks, depending on the concrete physical situations. In the first works, the ground state and finite temperature [21–23] mean values were considered directly in the infinite volume limit. An extension to the finite volume ground state was given in [24]. Excited states (with arbitrary Bethe root distributions) were later treated in [25], and finally, an extension to arbitrary finite volume excited states was given in [26].

As an alternative method towards factorized correlations, functional relations for reduced density matrices were considered in [27], and later for higher spin versions of the XXZ chain in [28,29]. Higher rank models were considered in [30–32]. The work [33] raises the question of whether all correlation functions of higher rank models can be expressed in a simple factorized form, and [34] treated factorized correlations in the XYZ model.

A new contribution to the theory was given in the works [35–37], where a connection was established to Generalized Hydrodynamics (GHD), a theory describing the large-scale transport properties of integrable models [38,39]. The papers [35–37] considered the mean values of current operators (including the so-called generalized current operators), which describe the flow of conserved charges during real-time evolution. It was found that these mean values are exceptionally simple: they are given simply by the Taylor coefficients of the so-called $\omega$ function. In the finite volume case, these building blocks can be expressed in a very simple way: they involve the one-particle charges corresponding to the Bethe roots and a single copy of the inverse of the so-called Gaudin matrix. The final expression has a semi-classical interpretation [35,40]: the currents of the conserved charges are given by a sum over the one-particle charge eigenvalues, multiplied by a certain effective velocity describing particle propagation in the background of the other particles. This simple result demonstrates the completely elastic and factorized scattering characteristic of integrable models, and it underlies the formulation of Generalized Hydrodynamics. The result for the current mean values was later extended to the XYZ model in [41].

It is then very natural to ask whether some of the above results about the currents and the factorized correlators can be worked out also for other integrable quantum spin chains. The algebraic construction of [37] appears rather general, but so far it has been applied only to the XXZ and XYZ spin chains. However, there is motivation to study other types of models as well.

As examples, we mention the integrable quantum spin chains acting on constrained Hilbert spaces with the Rydberg blockade [42]. The Hamiltonians of these models are related to the Restricted Solid On Solid (RSOS) models of Andrews, Baxter, and Forrester [43,44], and they have been studied for example in [45,46]. Interest in these models also comes from the various studies on the so-called PXP model and its relatives, see [47–52]. The transport in these integrable models has been studied numerically in [51], but GHD has not yet been established. At the same time, short-range correlation functions in these models were treated in [9,53], although general formulas have not yet been found. A specific model in this family is the so-called "golden chain" introduced in [54]. It is a special point in the integrable family where the Hamiltonian density satisfies the Temperley-Lieb algebra.

The goal of this work is to use the connections to the XXZ chain to find exact correlation functions in the golden chain and other Temperley-Lieb models. Such an approach cannot yield the factorized correlation functions of all short-range operators, but it is natural to expect exact relations for correlators built from the Temperley-Lieb algebra. This is the task that we set ourselves in this work. Apart from the golden chain, we will consider the $Q$-state Potts model and also the so-called trace representation of the Temperley-Lieb algebra, which appeared recently in the study of Hilbert space fragmentation [55].

In the main text, we formulate the conjecture that for operators constructed in terms of the Temperley-Lieb algebra, the factorization of the mean values works in essentially the same way in all representations of the Temperley-Lieb algebra, and only minor modifications need to be added. This seems to hold for all states with singlet eigenvalues of commuting set of transfer matrices. Thus, we claim that both the algebraic part and the physical part of the construction will be essentially the same. In particular, we conjecture that the physical part can be computed in finite volume using the formulas first derived in [26]. We test our conjectures in several representations. We also develop a method to compute generalized current operators from local charges, and afterward, we present a few concrete generalized current operators expressed via the Temperley-Lieb generators. These concrete formulas are then used to check our conjectures. We should note that in [56], explicit and exact formulas were found for all local conserved charges in the Temperley-Lieb models. However, we use a different basis of charges and, therefore, do not directly use the results of [56].

This paper consists of the following Sections: In Section 2 we review the Temperley-Lieb algebra and its several representations and explain the decomposition of the spectrum. In Section 3, we discuss the local charges in the Temperley-Lieb algebra and derive a way to construct the current operators. Section 4 gives a review of the factorization of correlation functions in the XXZ chain. In Section 5, we formulate our main conjectures and give some explicit formulas for short-range correlators. Section 6 contains our conclusions. Finally, some examples of the expressions for the local charges and currents in the Temperley-Lieb algebra are given in Appendix A, and some numerical checks are presented in Appendix B.

## 2 Temperley-Lieb algebra and its representations

The Temperley-Lieb (TL) algebra is defined as follows. There are generators $e_j$ with index $j = 1, \ldots, L-1$, which satisfy the relations

$$e_j^2 = d e_j, \qquad e_j e_{j\pm1} e_j = e_j, \qquad [e_j, e_k] = 0, \quad \text{for} \quad |j-k| > 1. \tag{1}$$

Here, $d \in \mathbb{C}$ is a fixed parameter of the algebra, which is also commonly parametrized as

$$d = \mathfrak{q} + \mathfrak{q}^{-1} = 2\cos\gamma, \qquad \mathfrak{q} = e^{i\gamma}. \tag{2}$$

Here, we are interested in the case of periodic boundary conditions, where another generator $e_L$ is introduced, satisfying [5]:

$$e_L^2 = d e_L\,, \quad e_L e_b e_L = e_L\,, \quad e_b e_L e_b = e_b\,, \quad [e_L, e_j] = 0\,, \tag{3}$$

where $b = 1, L-1$ and $j \neq 1, L-1$. The TL Hamiltonian with periodic boundary conditions is then defined as

$$H = \sum_{j=1}^{L} e_j\,. \tag{4}$$

The Temperley-Lieb algebra has multiple representations that are of physical interest. In particular, we will be interested in representations such as the XXZ spin chain, the quantum Potts chain, the RSOS representations (also known as anyon chains), and the so-called "trace representation".

A crucial fact is that since the Hamiltonian (4) is part of the (periodic) TL algebra, its spectrum in a given representation is entirely determined by the way the latter decomposes as a sum of irreducible representations. Starting from a given model, an important task is, therefore, to see how its Hilbert space decomposes as a direct sum of the TL irreducible representations.

Below, we discuss a few key statements about the representation theory of the TL algebra, and concrete representations will be considered in the following sections.

## 2.1 Irreducible representations of the (periodic) TL algebra

The TL algebra (1) can be viewed as an algebra of diagrams acting on $L$ vertical strands, by assigning to the generators $e_i$ the following graphical representation [57]

$$e_i = \bigg|\ \bigg|\ \bigg|\ \cdots\ \bigg|\ \underset{i\ \ i+1}{\overset{\smile}{\frown}}\ \bigg|\ \cdots\ \bigg|\ \underset{L}{\bigg|} \tag{5}$$

Multiplication of two generators corresponds to stacking the corresponding diagrams on top of each other, and the rules (1) translate into the fact that diagrams are identified modulo smooth stretching of strands, or removing of closed loops at the cost of a multiplicative scalar factor $d$.

A natural basis for representations of the TL algebra is constructed in terms of "reduced states", obtained by cutting in half horizontally all possible diagrams formed by products of TL generators. Those are constituted of a set of non-intersecting arcs joining pairs of strands, and "through lines", or "strings", which propagate vertically without the possibility of being contracted with one another by the action of TL generators. Irreducible representations are characterized by their number $2j$ of through-lines (where $0 \leq j \leq L/2$ is an integer when $L$ is even, a half-integer when $L$ is odd), and denoted as $\mathcal{W}_j[L]$. For instance, for $L = 4$ there are three irreducible modules with respective basis states

$$\begin{aligned}
\mathcal{W}_0[4] &= \{\cup\ \cup, \overset{\smile}{\smile}\}\,, \\
\mathcal{W}_1[4] &= \{|\ |\ \cup, \cup\ |\ |, |\ \cup\ |\}\,, \\
\mathcal{W}_2[4] &= \{|\ |\ |\ |\}\,.
\end{aligned} \tag{6}$$

Turning to periodic boundary conditions by introducing the generator $e_L \equiv e_0$, the TL algebra becomes infinite-dimensional, due to the possibility for through-lines to wind an arbitrary number of times around the cylinder, as well as, in the sector with no through-lines, the possibility for an arbitrary number of non-contractible loops wrapping around the cylinder.

Finite-dimensional quotients can be recovered by allowing to undo the winding of through-lines at the price of multiplication by a complex factor $z^{\pm 1}$ (one such factor per through-line per turn around the cylinder), and to eliminate non-contractible loops at the price of multiplication by a factor $z + z^{-1}$ [58,59]. The corresponding irreducible representations are denoted by $\mathcal{W}_{j,z}[L]$, and generically have more states than the open TL ones. For instance, for $L = 4$, basis states for the representations with $j \neq 0$ are

$$\mathcal{W}_{1,z}[4] = \{\mid\mid\cup,\cup\mid\mid,\mid\cup\mid,\cup\mid\mid\cup\},$$
$$\mathcal{W}_{2,z}[4] = \{\mid\mid\mid\mid\}. \tag{7}$$

For $j = 0$, we have to distinguish arcs which connect two strands by going one way or the other around the cylinder, therefore, a basis is

$$\mathcal{W}_{0,z}[4] = \{\cup\ \cup,\underset{\smile}{\cup},\cup\ \cup,\cup\ \cup,\underset{\smile}{\cup},\underset{\smile}{\cup}\}, \tag{8}$$

where an arc marked by a dot means that it goes around the periodic boundary conditions of the cylinder. All these representations have dimension

$$\dim \mathcal{W}_{j,z} = \binom{L}{L/2 - j}, \tag{9}$$

irrespectively of $z$.

These representations are irreducible for generic $\mathfrak{q}$ and $z$. For specific cases, however ($\mathfrak{q}$ equal to a root of unity or $z$ equal to some integer power of $\mathfrak{q}$), they become reducible but indecomposable, as a result of the TL algebra becoming non-semisimple (similar conclusions hold for the open case at root of unity). We refer to the existing literature [57, 60, 61] for a more detailed exposition.

## 2.2 The XXZ model

In this case, the generators act on the Hilbert space of a spin-1/2 chain, and we have

$$e_j = h_{j,j+1}^{\text{XXZ}}, \tag{10}$$

where $h_{j,j+1}^{\text{XXZ}}$ is a two-site operator given by [62]

$$h_{j,j+1}^{\text{XXZ}} = -\frac{1}{2}\Big[2e^{i\phi/L}\sigma_j^+\sigma_{j+1}^- + 2e^{-i\phi/L}\sigma_j^-\sigma_{j+1}^+ + \cos\gamma\big(\sigma_j^z\sigma_{j+1}^z - 1\big) + i\sin\gamma\big(\sigma_j^z - \sigma_{j+1}^z\big)\Big], \tag{11}$$

where $\gamma$ is the parameter introduced above (see eq. (2)), in practice, restricted to be real or pure imaginary. It is related to the so-called anisotropy parameter $\Delta$ of the XXZ chain by $\Delta = \cos\gamma$. Furthermore, $\phi \in \mathbb{C}$ is the so-called twist parameter. In our representation, we chose to apply a homogeneous distribution of the twist and focus on the periodic case. Note that the normalization of the Hamiltonian density is now different from the usual one: now it includes an overall factor of $-1/2$ so that $h_{j,j+1}^{\text{XXZ}}$ satisfies the Temperley-Lieb algebra.

The XXZ model can be solved by the Bethe Ansatz. Eigenstates are constructed using interacting spin waves, that are created on top of a reference state. The resulting eigenstates are characterized by a set of rapidities $\boldsymbol{\lambda}_N = \{\lambda_1,\ldots,\lambda_N\}$, which satisfy the Bethe equations

$$e^{i\phi}\left(\frac{\sinh(\lambda_j + i\gamma/2)}{\sinh(\lambda_j - i\gamma/2)}\right)^L \prod_{k \neq j}\frac{\sinh(\lambda_j - \lambda_k - i\gamma)}{\sinh(\lambda_j - \lambda_k + i\gamma)} = 1. \tag{12}$$

Here $\Delta = \cos(\gamma)$. If $|\Delta| < 1$ then $\gamma \in \mathbb{R}$ and the ground state configuration consists of real roots.

The energies of the states (eigenvalues of $H$ with the normalization given above) are given by

$$E = \sum_{j=1}^{N} \varepsilon(\lambda_j), \tag{13}$$

with

$$\varepsilon(\lambda) = \frac{\sin^2 \gamma}{\sinh(\lambda + i\gamma/2)\sinh(\lambda - i\gamma/2)}. \tag{14}$$

**Decomposition of the spectrum**   The Hilbert space can be split into sectors of fixed magnetization $S^z = \frac{1}{2}\sum_{i=1}^{L}\sigma_i^z$, and any operator written in terms of the TL generators (11) is block-diagonal in this decomposition. In fact, for generic $\gamma$ and $\phi$, we can identify each of these sectors with one of the irreducible representations of the periodic TL algebra. Recalling from Section 2.1 that those are indexed by a number $2j$ of "through lines" and by a "twist" parameter $z$, and denoted $\mathcal{W}_{j,z}$, we find that the XXZ sector of magnetization $S^z$ and twist $\phi$ identifies as the representation $\mathcal{W}_{j,z}$, with the following correspondence

$$j = |S^z|, \qquad z = e^{i\phi}. \tag{15}$$

Note in particular that sectors of opposite magnetization correspond to the same representation of TL and have the same spectrum.

Note that for a given TL representation, we can always find the same eigenenergy within a sector of the XXZ chain with the same parameter $d$ and with an appropriate twist up to the degeneracy [5].

## 2.3   The Potts model

The $Q$-states quantum Potts model is defined on a chain of $L/2$ sites carrying $Q$-dimensional spins. On each sites one defines matrices $X$ and $Z$, generalizing the Pauli matrices $\sigma^x$ and $\sigma^z$, which satisfy the following "$\mathbb{Z}_Q$ clock" algebra

$$X^\dagger = X^{Q-1}, \qquad Z^\dagger = Z^{Q-1}, \qquad X^Q = Z^Q = 1, \qquad XZ = \omega ZX, \tag{16}$$

where $\omega = e^{i\frac{2\pi}{Q}}$. A concrete representation can be obtained, for instance, by taking

$$Z = \begin{pmatrix} 1 & & & \\ & \omega & & \\ & & \ddots & \\ & & & \omega^{Q-1} \end{pmatrix}, \qquad X = \begin{pmatrix} 0 & 1 & & \\ & \ddots & \ddots & \\ & & \ddots & 1 \\ 1 & & & 0 \end{pmatrix}. \tag{17}$$

From there, the generators $e_1, \ldots e_L$ defined as [6]

$$e_{2j} = \frac{1}{\sqrt{Q}}\sum_{a=0}^{Q-1}\left(X_j^\dagger X_{j+1}\right)^a, \qquad e_{2j+1} = \frac{1}{\sqrt{Q}}\sum_{a=0}^{Q-1}Z_j^a, \tag{18}$$

satisfy the Temperley-Lieb algebra with $d = \sqrt{Q}$, and periodic boundary conditions. Furthermore, they are manifestly Hermitian operators.

**Decomposition of the spectrum**  As for the XXZ case, we can decompose the Hilbert space of the periodic Potts chain in terms of the representations $\mathcal{W}_{j,z}$, where we recall the correspondence (15) between the parameters $j, z$ and the magnetization and twist in the XXZ chain.

We start with the case $Q = 3$, which corresponds to $d = \sqrt{3} = 2\cos(\pi/6)$ and hence $\gamma = \frac{\pi}{6}$. At such "roots of unity" cases (namely, whenever $\mathfrak{q}$ is a root of unity), the TL algebra is known to be non-semisimple, which means that the representations $\mathcal{W}_{j,z}$ become reducible but indecomposable: they cannot be decomposed as a direct sum of irreducible representations. For our matters, we will not need to go into the details of this complicated subject, and we refer the interested reader to the existing literature [57, 59–61]. In practice, we should only stick to the observation that at the root of unity, the spectrum of the TL Hamiltonian (or more general operators built out of the TL algebra) in representation $\mathcal{W}_{j',z'}$ may arise as a subset of the spectrum in a larger $\mathcal{W}_{j,z}$, and we note $\mathcal{W}_{j,z}/\mathcal{W}_{j',z'}$ the remaining subspace. For $L = 4$, we find that the Potts chain Hilbert space decomposes as

$$\mathcal{H}^{Q=3} = \mathcal{W}_{0,-1} \oplus (\mathcal{W}_{0,\mathfrak{q}^2}/\mathcal{W}_{1,1}) \oplus \mathcal{W}_{2,1}\,. \tag{19}$$

Similar decompositions can be written for other system sizes $L$ (note that for $j = 2$ here, or $j = L/2$ in general, $\mathcal{W}_{L/2,z}$ is a one-dimensional representation where all TL generators cancel, so the precise value of the twist $z$ does not matter).

We now turn to $Q = 4$ and $Q = 5$, which correspond to $\gamma = 0$ and $\gamma = \arccos(\sqrt{5}/2) \simeq 0.48i$, respectively. Those are not "root of unity" points of the kind discussed above, however we shall still encounter quotients of the form $\mathcal{W}_{j,z}/\mathcal{W}_{j',z'}$. The reason is that, as explained in [8] (see also [58, 62]), even for generic $\mathfrak{q}$ representations $\mathcal{W}_{j,z}$ become reducible when $z = \mathfrak{q}^{2j+2k}$, where $k$ is some positive integer, and contain some irreducible submodule isomorphic to $\mathcal{W}_{j+k,\mathfrak{q}^{2j}}$. For $Q = 5$ we find accordingly, for $L = 4$

$$\mathcal{H}^{Q=5} = (\mathcal{W}_{0,\mathfrak{q}^2}/\mathcal{W}_{1,1}) \oplus 2\mathcal{W}_{0,-1} \oplus 11\mathcal{W}_{2,1}\,, \tag{20}$$

while for $Q = 4$, $L = 4$

$$\mathcal{H}^{Q=4} = \mathcal{W}_{0,1} \oplus (\mathcal{W}_{0,-1}/\mathcal{W}_{1,1}) \oplus \frac{1}{2}\mathcal{W}_{0,-1} \oplus 5\mathcal{W}_{2,1} \tag{21}$$

(in this last case, other subtleties arise, leading in particular to the $\frac{1}{2}$ factor which results from the fact that $\mathcal{W}_{0,-1}$ contains two copies of the same irreducible representation, but we shall not discuss these further here).

## 2.4  The golden chain

The Hamiltonian of the so-called golden chain was published in [54]. It is a special case of the family of Hamiltonians related to the RSOS models [45, 46].

In this case, the Hilbert space is constrained: in a volume $L$ it is spanned by the states of the computational basis, which do not have two neighboring down spins.

Let us define the local projectors

$$P_j = (1 + \sigma_j^z)/2\,, \qquad N_j = (1 - \sigma_j^z)/2\,. \tag{22}$$

Then, the constraint can be formalized as

$$N_j N_{j+1} = 0\,. \tag{23}$$

A representation of the Temperley-Lieb algebra is the following:

$$e_j = h_{j,j+1,j+2}\,, \tag{24}$$

where $h_{j,j+1,j+2}$ is a three-site operator acting on the constrained Hilbert space, given explicitly by

$$h_{j,j+1,j+2} = -\varphi \left[ (P_j + P_{j+2} - 1) - P_j P_{j+2} \left( \varphi^{-3/2} \sigma_{j+1}^x + \varphi^{-3} P_{j+1} + \varphi^{-2} + 1 \right) \right], \tag{25}$$

where $\varphi = (1 + \sqrt{5})/2 = 2\cos(\pi/5)$ is the golden ratio. The Temperley-Lieb parameter is $d = \varphi$.

**Decomposition of the spectrum** As for the XXZ and Potts representations, the RSOS Hilbert space can be decomposed in terms of the TL standard representations. We have here $\mathfrak{q} = e^{i\pi/5}$, again a root of unity, so similar comments to those made above for the Potts chain can be addressed here. We find, for $L = 4$:

$$\mathcal{H}^{\text{RSOS}} = (\mathcal{W}_{0,\mathfrak{q}^2}/\mathcal{W}_{1,1}) \oplus (\mathcal{W}_{0,\mathfrak{q}^4}/\mathcal{W}_{2,1}),$$

where again the correspondence with the XXZ parameters is encoded in (15).

## 2.5 The trace representation

In this representation, we are dealing with local Hilbert spaces $\mathbb{C}^d$ with $d \geq 2$, and the generators are given by

$$e_j = K_{j,j+1}, \tag{26}$$

where $K$ is the so-called trace operator, given explicitly by

$$K = \sum_{a,b=1}^{d} |aa\rangle\langle bb|. \tag{27}$$

In this case, the parameter of the Temperley-Lieb algebra is equal to the local dimension $d$ (more generally, representations where $d$ is a positive or negative integer can be constructed by using graded vector spaces [63]).

These models appeared recently in the study of Hilbert space fragmentation [55].

**Decomposition of the spectrum** We take $d = 3$ for the example. Here $\gamma = \arccos(3/2)$ is pure imaginary, and $\mathfrak{q} = e^{i\gamma}$ is not a root of unity. In terms of the modules $\mathcal{W}_{j,z}$, with again (15), we find for $L = 2$

$$\mathcal{H}^{d=3} = \mathcal{W}_{0,\mathfrak{q}^2} \oplus 7\mathcal{W}_{1,1}. \tag{28}$$

For $L = 4$, we find

$$\mathcal{H}^{d=3} = \mathcal{W}_{0,\mathfrak{q}^2} \oplus 7\mathcal{W}_{1,1} \oplus 47\mathcal{W}_{2,1}. \tag{29}$$

For $L = 6$, we find

$$\mathcal{H}^{d=3} = \mathcal{W}_{0,\mathfrak{q}^2} \oplus 7\mathcal{W}_{1,1} \oplus 27\mathcal{W}_{2,1} \oplus 20\mathcal{W}_{2,-1} \oplus 322\mathcal{W}_{3,1}. \tag{30}$$

# 3 Charges and currents in the Temperley-Lieb algebra

In this Section we discuss the family of conserved charges and their currents in the models related to the Temperley-Lieb algebra. To this order we first discuss the integrability properties, and afterward we turn to the charges and currents.

### 3.1 Integrability

The models defined by (4) are integrable, both in the periodic case and in the open case (where the additional generator $e_L$ is absent in (4)). The Hamiltonians can be embedded into a family of commuting transfer matrices. It is possible, by going to a specific representation, to construct commuting transfer matrices in finite volume with periodic or open boundary conditions. For the XXZ representation such transfer matrices are related to the six-vertex model [64]; in this case we give concrete formulas in the Appendix B.1. For the golden chain the transfer matrices are related to integrable RSOS models [65]; for the Potts representation, they are based on the Star-Triangle Relation [66,67]. However we do not know of a representation-independent formulation in finite volume, and will therefore restrict to a definition in infinite volume, valid for any representation.

First, we define

$$\check{R}_j(u) = 1 + u e_j, \tag{31}$$

where $u \in \mathbb{C}$ is a spectral parameter. Then, the formal definition of the transfer matrices can be given as

$$t(u) = \prod_j \check{R}_j(u). \tag{32}$$

For the ordering of the operators, we choose a convention that operators with lower indices act first. These operators should be regarded as a formal power series in $u$. Their finite volume counterparts can also be specified.

It can be shown by formal manipulation that

$$[t(u), t(v)] = 0. \tag{33}$$

Let us now define the operators with $k > 1$

$$A_k = (\partial_u)^{k-1} \log(t(u))\big|_{u=0}. \tag{34}$$

We have

$$A_2 = H. \tag{35}$$

It follows from (33) that the $A_k$ form a commuting family, and their construction ensures that they are extensive operators with a short-range operator density.

The alternative to the transfer matrix is the boost operator formalism [68–70]. This method for obtaining the charges is also limited to the infinite volume case. We define formally

$$\mathcal{B} = \sum_{j=-\infty}^{\infty} j e_j, \tag{36}$$

and define a series of charges via the formal rule

$$Q_{k+1} = [Q_k, \mathcal{B}], \tag{37}$$

with the initial condition

$$Q_2 = H. \tag{38}$$

These charges are linear combinations of the $A_k$ given by (34). The definition (37) does not yield Hermitian charges, in fact with this definition every second charge is anti-Hermitian. However, we use this convention because it is convenient for our purposes.

The Hamiltonians (4) also enjoy translational invariance, either in finite volume with periodic boundary conditions, or formally in the infinite volume case. However, there is a subtlety in connection with the translations.

For a given model, let us define $\mathcal{U}$ as the one-site translation operator acting on the physical Hilbert space. In all cases we have

$$[\mathcal{U}, H] = 0. \tag{39}$$

Furthermore, $\mathcal{U}$ also commutes with the higher charges. It is then very natural to consider the simultaneous diagonalization of $\mathcal{U}$ and the extensive local charges. The Bethe states will be eigenvectors of $\mathcal{U}$ as well, therefore $\mathcal{U}$ is the natural translational symmetry of both the charges and the states.

However, the situation is more delicate in the Potts model. In that case, we can also define the operator $\mathcal{U}$ based on its action in real space, but then $\mathcal{U}$ will shift the Temperley-Lieb generators by two indices, due to the staggering introduced in (18). On the level of operators, we have an additional symmetry $\mathcal{V}$, which shifts the indices of the operators by one. However, on the level of the Hilbert space, the operation $\mathcal{V}$ is seen as a duality, and it is not guaranteed that the Bethe states will be eigenvectors of $\mathcal{V}$.

## 3.2 Local charges and currents in Temperley-Lieb models

The charges introduced above are extensive and are expressed as

$$Q_\alpha = \sum_j q_\alpha(j), \tag{40}$$

where $q_\alpha(j)$ is a short-range operator density which can be expressed using the generators of the Temperley-Lieb algebra.

We choose our conventions such that $q_\alpha(j)$ spans $\alpha$ sites in the XXZ representation, and we choose $q_2(j) = e_j$, which also implies that $Q_2 = H$ in our conventions.

The first non-trivial charge above the Hamiltonian is

$$q_3(j) = e_j e_{j+1} - e_{j+1} e_j. \tag{41}$$

Further examples are found in the Appendix A.

The general explicit expressions for the local charges of the Temperley-Lieb models were derived by Nienhuis and Huijgen [56]. However, these expressions are some linear combination of the charges obtained from expanding the transfer matrix constructed from the usual 6-vertex R-matrix in the XXZ representation, or obtained through the boost operation. Therefore, the concrete formulas of [56] coincide neither with our $A_k$ nor with our $Q_k$.

The current operators $J_\alpha(x)$ are defined through the continuity equations

$$[H, q_\alpha(x)] = J_\alpha(x) - J_\alpha(x+1). \tag{42}$$

We also introduce the generalized currents $J_{\alpha,\beta}(x)$ [35, 37] that describe the flow of $q_\alpha(x)$ under the time evolution generated $Q_\beta$ by

$$\left[Q_\beta, q_\alpha(x)\right] = J_{\alpha,\beta}(x) - J_{\alpha,\beta}(x+1). \tag{43}$$

The locality of the charge densities and the global relation $[Q_\alpha, Q_\beta] = 0$ imply that the operator equation (43) can always be solved with a short-range operator $J_{\alpha,\beta}(x)$.

The definition (43) holds for $\alpha, \beta \geq 2$, but it will turn out convenient to extend it to $\beta = 1$ by setting

$$J_{\alpha,1}(x) \equiv q_\alpha(x). \tag{44}$$

### 3.3 Construction of the current operators

Below, we demonstrate the procedure for constructing the expressions of current operators and generalized current operators from the corresponding expressions of local conserved quantities. The construction of the current operators was originally derived for the spin-1/2 XYZ chain [71]. Here, we generalize this result of [71] to the construction of the generalized current operators. Notably, this method applies to general quantum integrable spin chains of local Hamiltonian. We first show the general procedure, which does not involve the Temperley-Lieb algebra. Afterwards, we apply the result to the Temperley-Lieb Hamiltonian.

We start by reviewing the procedure for constructing the current operators [71]. In the following, we refer to an operator that acts on the $x$-th site and also locally on sites beyond the $x$-th site as "starting from the $x$-th site". We assume $q_\alpha(x)$ is starting from $x$-th site. Since the Hamiltonian is a two-site operator, the left-hand side of (42) is constructed from the operator starting from the $(x-1)$-th, $x$-th, and $(x+1)$-th sites, which we denote by $F_\alpha^{-1}(x-1)$, $F_\alpha^0(x)$, and $F_\alpha^1(x+1)$, respectively:

$$[H, q_\alpha(x)] = F_\alpha^{-1}(x-1) + F_\alpha^0(x) + F_\alpha^1(x+1). \tag{45}$$

The current operator is given by

$$J_\alpha(x) = F_\alpha^{-1}(x-1) - F_\alpha^1(x). \tag{46}$$

The proof of (46) is as follows: substituting (46) to the RHS of (42), we have

$$J_\alpha(x) - J_\alpha(x+1) = [H, q_\alpha(x)] - \delta_\alpha(x), \tag{47}$$

where we defined $\delta_\alpha(x) \equiv F_\alpha^{-1}(x) + F_\alpha^0(x) + F_\alpha^1(x)$. Summing over $x$ of (45), we have $0 = \sum_{x=1}^L \delta_\alpha(x)$. Since $\{\delta_\alpha(x)\}_{x=1,2,\dots,L}$ are mutually linear independent, each $\delta_\alpha(x)$ should be zero itself. Now, we have proved (46) satisfies the continuity equation (42).

Concrete examples for current operators are found in the Appendix A.

### 3.4 Construction of the generalized current operators

We generalize the result for the current operator [71] to the generalized current operator. Since $Q_\beta$ is a $\beta$-site operator, the left-hand side of (42) is written as

$$\left[Q_\beta, q_\alpha(x)\right] = \sum_{y=-(\beta-1)}^{\beta-1} F_{\alpha,\beta}^y(x+y), \tag{48}$$

where $F_{\alpha,\beta}^y(x+y)$ is an operator starting from $x+y$-th site. By summing up the $x$ in (48) and in the same manner as the usual current operator case, we have

$$\sum_{y=-(\beta-1)}^{\beta-1} F_{\alpha,\beta}^y(x) = 0. \tag{49}$$

The generalized current operator is given by

$$J_{\alpha,\beta}(x) = \sum_{b=1}^{\beta-1} \sum_{y=1}^b \left[ F_{\alpha,\beta}^{-b}(x-y) - F_{\alpha,\beta}^b(x+y-1) \right]. \tag{50}$$

The $\beta = 2$ case of (50) recovers (46).

We prove (50) in the following. We define $E_\pm^y(x) \equiv \sum_{b=y}^{\beta-1} F_{\alpha,\beta}^{\pm b}(x)$, and we can write the generalized current as $J_{\alpha,\beta}(x) = \sum_{y=1}^{\beta-1}\left[E_-^y(x-y) - E_+^y(x+y-1)\right]$. Substituting (50) to the RHS of (43), we have

$$
\begin{aligned}
J_{\alpha,\beta}(x) - J_{\alpha,\beta}(x+1) &= \sum_{y=1}^{\beta-1}\left[E_-^y(x-y) + E_+^y(x+y)\right] - \sum_{y=1}^{\beta-1}\left[E_-^y(x-y+1) + E_+^y(x+y-1)\right] \\
&= \sum_{\beta-1\geq|y|>0} F^y(x+y) + \sum_{y=1}^{\beta-2}\left[E_-^{y+1}(x-y) + E_+^{y+1}(x+y)\right] \\
&\quad - \sum_{y=1}^{\beta-1}\left[E_-^y(x-y+1) + E_+^y(x+y-1)\right] \\
&= \left[Q_\beta, q_\alpha(x)\right] - F_{\alpha,\beta}^0(x) - E_+^1(x) - E_-^1(x) \\
&= \left[Q_\beta, q_\alpha(x)\right] - \sum_{y=-\beta+1}^{\beta-1} F_{\alpha,\beta}^y(x) \\
&= \left[Q_\beta, q_\alpha(x)\right],
\end{aligned}
\tag{51}
$$

where in the second equality, we have used $E_\pm^y(x) = E_\pm^{y+1}(x) + F_{\alpha,\beta}^y(x)$ and $E_\pm^\beta(x) = 0$, and in the last equality we have used (49). Thus, we have proved (50) actually satisfies (43).

In principle, we can obtain the explicit expressions of the generalized currents by evaluating the commutator in the continuity equation and subsequently obtaining $F_{\alpha,\beta}^y(x)$ and using (50), provided that we have the explicit expressions of the local charges $Q_\alpha$.

The aforementioned index $x$ need not strictly denote a physical site index; the only requirement is that the operators with different indices are to be linearly independent. Thus, the above construction of the generalized currents is also applicable to the Potts representation where the indices of the Temperley-Lieb generators do not correspond to the physical site indices.

Concrete examples for generalized current operators in the Temperley-Lieb algebra are found in the Appendix A.

## 4 Factorized correlation functions in XXZ

### 4.1 Hidden Grassmann structure

The theory of the factorized correlation functions [13–20] concerns the mean values of local operators in the XXZ chain in a variety of physical situations. The goal is to compute mean values of the form

$$
\langle\Psi|\mathcal{O}|\Psi\rangle,
\tag{52}
$$

in finite volume or in the thermodynamic limit. The papers [13–20] considered the ground state and finite temperature situations in infinite volume. The ground state in finite volume was treated in [24], and an extension to arbitrary finite volume excited states was given in [26]. Excited states in thermodynamic limit were treated earlier in [25].

The results of the theory for the homogeneous limit can be summarized as follows. The statements below hold for all the physical situations mentioned above, thus also for the finite volume excited states considered in [26].

The mean values can be expressed as a sum of products of certain building blocks, which are the Taylor expansion coefficients of a few number of functions of one and two variables.

For example, in the XXZ spin chain the mean values of spin reflection invariant operators are expressed using the Taylor coefficients of just two functions $\omega(x,y)$ and $\omega'(x,y)$, both of which are symmetric with respect to the exchange of the two variables. The construction consists of the *algebraic part* and the *physical part*. The algebraic part describes the expressions of the mean values of a given operator as a combination of the building blocks, and this computation is independent of the physical situation. The physical part gives concrete values to the functions involved, depending on the physical situation considered.

Simple examples of short-range correlators are:

$$
\begin{aligned}
\left\langle \sigma_1^z \sigma_2^z \right\rangle &= \coth(\eta)\omega_{0,0} + W_{1,0}\,, \\
\left\langle \sigma_1^x \sigma_2^x \right\rangle &= -\frac{\omega_{0,0}}{2\sinh(\eta)} - \frac{\cosh(\eta)}{2}W_{1,0}\,, \\
\left\langle \sigma_1^z \sigma_3^z \right\rangle &= 2\coth(2\eta)\omega_{0,0} + W_{1,0} + \tanh(\eta)\frac{\omega_{2,0}-2\omega_{1,1}}{4} - \frac{\sinh^2(\eta)}{4}W_{2,1}\,, \\
\left\langle \sigma_1^x \sigma_3^x \right\rangle &= -\frac{1}{\sinh(2\eta)}\omega_{0,0} - \frac{\cosh(2\eta)}{2}W_{1,0} - \tanh(\eta)\cosh(2\eta)\frac{\omega_{2,0}-2\omega_{1,1}}{8} + \sinh^2(\eta)\frac{W_{2,1}}{8}\,.
\end{aligned}
\tag{53}
$$

Here

$$
\omega_{a,b} = \omega_{b,a} = (\partial_x)^a(\partial_y)^b \omega(x,y)\big|_{x,y=0}\,,
\tag{54}
$$

and the function $\omega(x,y)$ coincides with the one defined in [21, 22], and $W_{a,b}$ are given similarly by the coefficients of the function $W(x,y) = \omega'(x,y)/\eta$, where $\omega'(x,y)$ is defined in [21, 22] and $\Delta = \cosh(\eta)$. The concrete values of these functions in the finite temperature ensembles were given in [21, 22], whereas for finite volume excited states they were computed in [26].

For operators which are invariant under the action of the quantum group $U_q(sl(2))$ the mean values involve only the function $\omega$ [18, 72]. For the closely related $SU(2)$-symmetric case see [20, 24, 27, 73]. In the XXZ chain the operators invariant under the quantum group are generated by the Temperley-Lieb algebra. Therefore, the works [18, 72] compute those mean values which we are interested in, specifically in the XXZ chain.

## 4.2 Specific results for mean value of the current operators

The key result of the works [35–37] is that the mean values of the generalized currents are given by the coefficients of the $\omega$ function. This was proven in [35–37] by realizing that the formulas for the current mean values are identical to the formulas for the $\omega$ function found originally in [26].

Let us now summarize these results. For our purposes it is convenient to define a new function $\psi(x,y) = \psi(y,x)$ which is related to $\omega$ via

$$
\psi(x,y) = -i\sin(\gamma)\left[\frac{1}{2}\omega(i\sin(\gamma)x, i\sin(\gamma)y) + \frac{i}{4}K(i\sin(\gamma)(x-y))\right],
\tag{55}
$$

where

$$
K(u) = \frac{\sin(2\gamma)}{\sinh(u+i\gamma)\sinh(u-i\gamma)}\,.
\tag{56}
$$

Similarly to (54) we define

$$
\psi_{a,b} = \psi_{b,a} = (\partial_x)^a(\partial_y)^b \psi(x,y)\big|_{x,y=0}\,.
\tag{57}
$$

They are related to the coefficients $\omega_{a,b}$ via

$$
\psi_{a,b} = -\left[\frac{1}{2}\omega_{a,b} + (-1)^a\frac{i}{4}\left((\partial_u)^{a+b}K(u)\,|_{u=0}\right)\right]\sinh^{a+b+1}(i\gamma).
\tag{58}
$$

The main result of [35–37] is that for any Bethe state $|\boldsymbol{\lambda}_N\rangle$ given by the set of Bethe roots $\boldsymbol{\lambda}_N = \{\lambda_1, \ldots, \lambda_N\}$, the current mean values are given by

$$\langle\boldsymbol{\lambda}_N|J_{\alpha,\beta}|\boldsymbol{\lambda}_N\rangle = \psi_{\alpha-2,\beta-1}, \tag{59}$$

where $\psi(x, y)$ is now defined by the $\omega$ function for the Bethe state $|\boldsymbol{\lambda}_N\rangle$, and it is written using the Bethe roots by

$$\psi(x, y) = \mathbf{h}(i\sin(\gamma)x) \cdot G^{-1} \cdot \mathbf{h}(i\sin(\gamma)y) \times (-\sin(\gamma)), \tag{60}$$

where $\mathbf{h}(x)$ is a parameter dependent vector of length $N$ with elements $\mathbf{h}_j(x) = h(\lambda_j - x)$ with $h(\lambda)$ given by

$$h(\lambda) = \coth(\lambda - i\gamma/2) - \coth(\lambda + i\gamma/2), \tag{61}$$

and $G$ is the Gaudin matrix, defined as

$$G_{jk} = \delta_{jk}\left(L\frac{\sin(\gamma)}{\sinh(\lambda_j + i\gamma/2)\sinh(\lambda_j - i\gamma/2)} - \sum_{l=1}^{N}K(\lambda_{jl})\right) + K(\lambda_{jk}), \tag{62}$$

where we denote $\lambda_{jk} \equiv \lambda_j - \lambda_k$.

In the specific case of the charge densities we get

$$\langle\{\lambda\}_N|q_\alpha|\{\lambda\}_N\rangle = \psi_{0,\alpha-2}. \tag{63}$$

Note that the quantities $\psi_{\alpha,\beta}$ are symmetric with respect to $\alpha, \beta$, but the current mean values involve certain shifts in the indices. These shifts appear due to our definitions of the charges and currents. The symmetry of the $\psi_{\alpha,\beta}$ leads to the symmetry

$$\langle\boldsymbol{\lambda}_N|J_{\alpha,\beta}|\boldsymbol{\lambda}_N\rangle = \langle\boldsymbol{\lambda}_N|J_{\beta+1,\alpha-1}|\boldsymbol{\lambda}_N\rangle, \tag{64}$$

for the mean values. This equation does not hold on the level of the operators, only on the level of the mean values.

## 5 Correlation functions

In this section, we formulate two conjectures about the correlation functions for the quantum integrable models, which are representations of the (affine) TL algebra (4).

We are interested in correlation functions of the form

$$\frac{\langle\Psi_L|\mathcal{O}|\Psi_R\rangle}{\langle\Psi_L|\Psi_R\rangle}. \tag{65}$$

Here, $|\Psi_R\rangle$ is an eigenvector of the Temperley-Lieb Hamiltonian in a selected representation. For the correlation functions, we will consider only the periodic case. Furthermore, we will focus on states which are singlets of the *transfer matrix*. The vector $\langle\Psi_L|$ is the left eigenvector of the transfer matrices corresponding to the same eigenvalue.

The operator $\mathcal{O}$ is chosen to be a word from the Temperley-Lieb algebra. We will focus on *short-range operators*, which include products of Temperley-Lieb generators with indices close to each other.

## 5.1 Main results and numerical checks

We first introduce the linear map $\mathcal{T}$ over the TL algebra. This map generates a shift-invariant operator, such that the result is translationally invariant with respect to the Temperley-Lieb indices. More concretely, the action of $\mathcal{T}$ is defined by:

$$\mathcal{T}[e_{i_1}e_{i_2}\cdots e_{i_m}] \equiv \frac{1}{L}\sum_{j=1}^{L} e_{i_1+j}e_{i_2+j}\cdots e_{i_m+j}. \tag{66}$$

In many representations, $\mathcal{T}$ generates operators that are invariant with respect to a shift in the physical sites. However, in the case of the Potts model, we are dealing with the invariance with respect to shifts in the Temperley-Lieb indices, which is not identical to the physical shifts.

In the following, we assume $|E\rangle$ is an eigenstate of a Hamiltonian of TL representation with an eigenenergy $E$. We assume that $|E\rangle$ is a singlet of the commuting set of charges. We define the translational mean value $\langle\cdot\rangle \equiv \langle E|\mathcal{T}[\,\cdot\,]|E\rangle$.

Below, we present two conjectures, which we tested in multiple representations and various volumes and singlet eigenstates.

**Conjecture 1** *The translationally invariant mean values of the generalized current operators are given by*

$$\langle J_{\beta+2,\alpha+1}(j)\rangle = \psi_{\alpha,\beta} \ (\alpha,\beta \geq 0), \tag{67}$$

*where $\psi_{\alpha,\beta}$ are the Taylor coefficients of the function $\psi(x,y)$, which is expressed via the Bethe roots in eq. (60), corresponding to the eigenstate with the same value of local charges in the XXZ representation with the same TL parameter $d$, and appropriate twist $\phi$.*

The expression for $\psi_{\alpha,\beta}$ is the same in all different representations with the same TL parameters $d$.

**Conjecture 2** *In eigenstates $|E\rangle$ which are singlets, the translationally invariant mean values of operators generated by the TL algebra are universally factorized as:*

$$\langle e_{i_1}e_{i_2}\cdots e_{i_m}\rangle = \sum_{p=1}^{p_f}\sum_{\boldsymbol{\alpha},\boldsymbol{\beta}}\left(\prod_{j=1}^{p}\psi_{\alpha_j,\beta_j}\right)C_{\boldsymbol{\alpha},\boldsymbol{\beta}}(d), \tag{68}$$

*where $\boldsymbol{\alpha} = \{\alpha_1,\alpha_2,\ldots,\alpha_p\}$ and $\boldsymbol{\beta} = \{\beta_1,\beta_2,\ldots,\beta_p\}$ satisfies $0 \leq \alpha_n \leq \beta_n \leq M$ and $\alpha_n \leq \alpha_{n+1}$ and $\beta_n \leq \beta_{n+1}$, and $M$ and $p_f$ are some positive integers. The coefficients $C_{\boldsymbol{\alpha},\boldsymbol{\beta}}(d)$ are independent of the choice of the eigenstate $|E\rangle$ and the representation of the TL algebra. The information about our choice of the operator $e_{i_1}\ldots e_{i_m}$ is encoded in $C_{\boldsymbol{\alpha},\boldsymbol{\beta}}(d)$. They depend only on the parameter $d$ of the TL algebra. This implies that all the $C_{\boldsymbol{\alpha},\boldsymbol{\beta}}(d)$ can be found in the concrete example of the XXZ spin chain, and they can be extracted from the existing works [18, 22, 72].*

We stress that the conjectures refer to the shift-invariant mean values generated by the operator $\mathcal{T}$. In many models, it is not necessary to introduce $\mathcal{T}$, because all mean values are shift-invariant. However, in the Potts model, there is a distinction between translational invariance and shift invariance, and while all mean values are translationally invariant (with respect to physical translations), they are not shift-invariant. We observed that the conjectures hold only for the shift-invariant combinations.

We computed several examples of factorized formulas, using the results of [22]. These are presented in the following. Furthermore, in Subsection 5.2 below we consider special combinations of charges and currents, whose mean values are always bi-linear in $\psi_{a,b}$.

The simplest short-range correlation function (involving at least two TL generators) is

$$\langle e_1 e_2 \rangle = \frac{1}{d}\left(\frac{1}{2}\psi_{0,2} - 2\psi_{0,0} - \psi_{1,1}\right) + \frac{1}{2}\psi_{0,1}. \tag{69}$$

In this case, there is no actual factorization happening because the decomposition above works on the level of the operators, guaranteed by the operator identity

$$e_1 e_2 = \frac{1}{d}\left\{\frac{1}{2}q_4(1) - 2q_2(2) - J_{3,2}(2)\right\} + \frac{1}{2}q_3(1). \tag{70}$$

Actual factorization is observed in more complicated cases. For example

$$\langle e_1 e_3 \rangle = \frac{1}{d(d^2-1)}\left[(d^2-4)\psi_{0,0} + 2\psi_{0,2} + \frac{d^2-40}{12}\psi_{1,1} + \frac{1}{6}\psi_{1,3} - \frac{1}{4}\psi_{2,2}\right]$$
$$+ \frac{1}{d^2-1}\left[\frac{d^2-28}{12}(\psi_{0,1}^2 - \psi_{0,0}\psi_{1,1}) - \frac{1}{4}(\psi_{0,2}^2 - \psi_{0,0}\psi_{2,2})\right.$$
$$\left. + \frac{1}{2}(\psi_{0,2}\psi_{1,1} - \psi_{0,1}\psi_{1,2}) + \frac{1}{6}(\psi_{0,1}\psi_{0,3} - \psi_{0,0}\psi_{1,3})\right]. \tag{71}$$

Furthermore

$$\langle e_1 e_2 e_3 + e_3 e_2 e_1 \rangle = \frac{1}{6d(d^2-1)}\left[5d^3\psi_{1,1} + 11d^2\left(\psi_{1,0}^2 - \psi_{0,0}\psi_{1,1}\right)\right.$$
$$+ d(36\psi_{0,0} + 34\psi_{1,1} - 24\psi_{2,0} + 3\psi_{2,2} - 2\psi_{3,1}) + 3\psi_{2,0}(\psi_{2,0} - 2\psi_{1,1})$$
$$\left. + 2\psi_{1,0}(8\psi_{1,0} + 3\psi_{2,1} - \psi_{3,0}) + \psi_{0,0}(-16\psi_{1,1} - 3\psi_{2,2} + 2\psi_{3,1})\right], \tag{72}$$

and

$$\langle e_1 e_4 \rangle = \frac{1}{d(d^2-1)(d^2-2)}\left[2(d^2-4)\psi_{0,0} - \frac{11d^2-92}{12}\psi_{0,2} + \frac{d^4+40d^2-392}{36}\psi_{1,1} + \frac{d^2+56}{36}\psi_{1,3}\right.$$
$$\left. - \frac{d^2+44}{24}\psi_{2,2} - \frac{1}{12}\psi_{0,4} + \frac{1}{24}\psi_{2,4} - \frac{1}{18}\psi_{3,3}\right]$$
$$+ \frac{1}{(d^2-1)(d^2-2)}\left[\frac{5d^4-28d^2-220}{18}\left(\psi_{0,1}^2 - \psi_{0,0}\psi_{1,1}\right)\right.$$
$$+ \frac{2d^2+16}{9}\left\{2(\psi_{0,1}\psi_{0,3} - \psi_{0,0}\psi_{1,3}) + 3(\psi_{0,0}\psi_{2,2} - \psi_{0,2}^2)\right\}$$
$$+ \frac{7d^4+226d^2+928}{144}\left(\psi_{0,2}\psi_{1,1} - \psi_{0,1}\psi_{1,2}\right)$$
$$+ \frac{5d^2+22}{144}\left\{\psi_{0,1}\psi_{1,4} - \psi_{1,1}\psi_{0,4} - 2\psi_{0,1}\psi_{2,3} + 6(\psi_{1,1}\psi_{2,2} - \psi_{1,2}^2)\right\}$$
$$+ \frac{10d^2+140}{144}\psi_{0,3}\psi_{1,2} + \frac{1}{6}\left(\psi_{0,2}\psi_{0,4} - \psi_{0,0}\psi_{2,4}\right) + \frac{2}{9}\left(\psi_{0,0}\psi_{3,3} - \psi_{0,3}^2\right) - \frac{2}{3}\psi_{0,2}\psi_{1,3}$$
$$+ \frac{1}{12}\left(\psi_{1,3}\psi_{2,2} - \psi_{1,2}\psi_{2,3}\right) + \frac{1}{18}\left(\psi_{1,1}\psi_{3,3} - \psi_{1,3}^2\right) + \frac{1}{24}\left(\psi_{1,2}\psi_{1,4} - \psi_{1,1}\psi_{2,4}\right)$$
$$\left. + \frac{1}{36}\left(\psi_{0,3}\psi_{2,3} - \psi_{0,2}\psi_{3,3}\right) + \frac{1}{48}\left(\psi_{0,2}\psi_{2,4} - \psi_{0,4}\psi_{2,2}\right) + \frac{1}{72}\left(\psi_{0,4}\psi_{1,3} - \psi_{0,3}\psi_{1,4}\right)\right]. \tag{73}$$

We numerically confirmed the above equations in the XXZ chain, the golden chain, the Potts chain with $2 \leq Q \leq 5$, and for the trace representation, in various finite volumes. A selection of concrete numerical data is presented in Appendix B. We note that the denominator of (73) diverges for $Q = 2$ Potts (Ising case), thus we checked the formula (73) works in the Potts chain for $2 < Q \leq 5$. However, we checked in the representations where $d$ can be tuned continuously that this divergence is compensated by a cancellation of the numerator as

$d^2 \to 2$. A similar phenomenon, which has also been observed in the works [9,53] occurs also for the above correlators as $d^2 \to 1$, and ensures that all correlation functions remain finite.

Here, we only consider the eigenstates which are singlets of the commuting set of charges, i.e., the eigenstates that can be distinguished by the values of the local charges and also the fundamental charges, such as the momentum operator. In the setups where we used to check Conjecture 1 and Conjecture 2 up to $L = 10$, almost all eigenstates are singlets, and we confirmed Conjecture 1 and Conjecture 2 holds for singlet eigenstates. For the case that q is a root of unity, we observed that non-singlet eigenstates can appear. In the case of non-singlet eigenstates, there remains the freedom to select a basis within the degenerate eigensubspace, and the value of the correlation function might vary depending on this choice. We leave the treatment for non-singlet eigenstates to future research.

## 5.2 Special bi-linear combinations

The work [74] found a special family of operators, whose mean values always factorize in integrable quantum spin chains. These operators are bi-linear combinations of (generalized) currents and charge densities. Now we also summarize these results here.

Let us introduce the operators $K_{\alpha,\beta,\gamma}(j,k)$, which depend on three indices $\alpha, \beta, \gamma$ and two coordinates (site indices) $j, k$:

$$K_{\alpha,\beta,\gamma}(j,k) = J_{\alpha,\gamma}(j)q_\beta(k) - q_\alpha(j)J_{\beta,\gamma}(k+1), \tag{74}$$

It can be shown that the mean values of these operators do not depend on $k$; the proof relies on the continuity equations. Afterwards, it can be shown that the mean values factorize, namely for the mean values in all eigenstates we have

$$\left\langle K_{\alpha,\beta,\gamma}(j,k) \right\rangle = \left\langle J_{\alpha,\gamma} \right\rangle \left\langle q_\beta \right\rangle - \left\langle q_\alpha \right\rangle \left\langle J_{\beta,\gamma} \right\rangle. \tag{75}$$

In the present conventions this means that

$$\left\langle K_{\alpha,\beta,\gamma}(j,k) \right\rangle = \psi_{\alpha-2,\gamma-1}\psi_{\beta-2,0} - \psi_{\alpha-2,0}\psi_{\beta-2,\gamma-1}. \tag{76}$$

These special operator combinations can be seen as the lattice version of the famous $T\bar{T}$-operator known in conformal field theories [75,76].

# 6 Conclusions and outlook

It has been known for many decades that the Temperley-Lieb algebra connects the spectra of different models, which are seen as different representations of the same algebra. However, no systematic study of correlation functions of Temperley-Lieb models existed before our work. The main conclusion of this work is that the TL algebra also connects the correlation functions, and that the existing results of the XXZ spin chain can be used to compute correlation functions in many other models. It is important that our results are not limited to the ground states, instead they can be applied to any state which is a singlet of the local charges in the Temperley-Lieb algebra.

We presented our results in the form of conjectures, an actual proof of which is desirable. A possible route to follow is to exploit the algebraic construction of the conserved current in integrable models [37] in order to derive algebraic relations relating the latter with local correlators. We plan to explore this direction in a future work.

At present, our conjectures hold for the translationally invariant mean values. In those cases when the translational invariance is broken by the representation of the Temperley-Lieb algebra (for example the Potts chain), we observed that the conjectures hold for the

translationally invariant averages. At the moment there is no theoretical explanation as to why the conjectures still work for the averages. This is a question that also deserves further study.

Also, it would be useful to investigate those states which are not singlets of the Temperley-Lieb algebra. In such cases it was observed that the mean values can depend on the choice of the basis within the degenerate sub-space. A more detailed investigation is beyond the scope of the present paper.

As a by-product of our computations, we found an apparently new way to construct the generalized current operators from the expressions for the local charges, see Sections 3.3-3.4. This result is independent of the Temperley-Lieb algebra, therefore it could be applied to the other integrable spin chains, and it might facilitate the search for factorized correlators in other circumstances.

## Acknowledgments

We are thankful to Hermann Boos, Frank Göhmann, Holger Frahm, Arthur Hutsalyuk, Andreas Klümper, Yuan Miao, Levente Pristyák and Fedor Smirnov for useful discussions.

**Funding information** K. F. was supported by FoPM, WINGS Program, JSR Fellowship, the University of Tokyo, and KAKENHI Grants No. JP21J20321 from the Japan Society for the Promotion of Science.

## A    Current operators of Temperley-Lieb Hamiltonian

Here, we provide examples of the current and generalized current operators. We use the conventions given in the main text; this convention corresponds to using the boost operator, or equivalently, the usual R-matrix in the 6-vertex model.

Examples of the lower-order charges are

$$q_4(j) = 2(e_{j+2}e_{j+1}e_j - e_{j+1}e_je_{j+2} - e_je_{j+2}e_{j+1} + e_je_{j+1}e_{j+2}) + de_{j+1}e_j + de_je_{j+1}, \tag{A.1}$$

$$\begin{aligned} q_5(j) = {}&6(e_je_{j+1}e_{j+2}e_{j+3} - e_{j+3}e_{j+2}e_{j+1}e_j + e_{j+2}e_{j+1}e_je_{j+3} + e_{j+1}e_je_{j+3}e_{j+2} + e_je_{j+3}e_{j+2}e_{j+1} \\ &- e_{j+1}e_je_{j+2}e_{j+3} - e_je_{j+2}e_{j+1}e_{j+3} - e_je_{j+1}e_{j+3}e_{j+2} - de_{j+2}e_{j+1}e_j + de_je_{j+1}e_{j+2}) \\ &+ (2+d^2)(e_je_{j+1} - e_{j+1}e_j), \end{aligned} \tag{A.2}$$

$$\begin{aligned} q_6(j) = {}&24(e_{j+4}e_{j+3}e_{j+2}e_{j+1}e_j - e_{j+3}e_{j+2}e_{j+1}e_je_{j+4} - e_{j+2}e_{j+1}e_je_{j+4}e_{j+3} - e_{j+1}e_je_{j+4}e_{j+3}e_{j+2} \\ &- e_je_{j+4}e_{j+3}e_{j+2}e_{j+1} + e_{j+2}e_{j+1}e_je_{j+3}e_{j+4} + e_{j+1}e_je_{j+3}e_{j+2}e_{j+4} + e_{j+1}e_je_{j+2}e_{j+4}e_{j+3} \\ &+ e_je_{j+3}e_{j+2}e_{j+1}e_{j+4} + e_je_{j+2}e_{j+1}e_{j+4}e_{j+3} + e_je_{j+1}e_{j+4}e_{j+3}e_{j+2} - e_{j+1}e_je_{j+2}e_{j+3}e_{j+4} \\ &- e_je_{j+2}e_{j+1}e_{j+3}e_{j+4} - e_je_{j+1}e_{j+3}e_{j+2}e_{j+4} - e_je_{j+1}e_{j+2}e_{j+4}e_{j+3} + e_je_{j+1}e_{j+2}e_{j+3}e_{j+4}) \\ &+ 36d(e_{j+3}e_{j+2}e_{j+1}e_j + e_je_{j+1}e_{j+2}e_{j+3}) - 12d(e_{j+2}e_{j+1}e_je_{j+3} + e_{j+1}e_je_{j+3}e_{j+2} \\ &+ e_je_{j+3}e_{j+2}e_{j+1} + e_{j+1}e_je_{j+2}e_{j+3} + e_je_{j+2}e_{j+1}e_{j+3} + e_je_{j+1}e_{j+3}e_{j+2}) \\ &+ (16+14d^2)(e_{j+2}e_{j+1}e_j + e_je_{j+1}e_{j+2}) + (-16-2d^2)(e_{j+1}e_je_{j+2} + e_je_{j+2}e_{j+1}) \\ &+ (8d+d^3)(e_{j+1}e_j + e_je_{j+1}) - 24de_je_{j+2}. \end{aligned} \tag{A.3}$$

We give a few examples of the corresponding current operators:

$$J_{3,2}(j) = e_{j+1}e_j e_{j-1} - e_j e_{j-1}e_{j+1} - e_{j-1}e_{j+1}e_j + e_{j-1}e_j e_{j+1} - 2e_j, \tag{A.4}$$

$$\begin{aligned}J_{3,3}(j) = &-e_{j+2}e_{j+1}e_j e_{j-1} - e_{j+1}e_j e_{j-1}e_{j-2} + e_{j+1}e_j e_{j-1}e_{j+2} + e_j e_{j-1}e_{j+2}e_{j+1} + e_j e_{j-1}e_{j-2}e_{j+1}\\
&+ e_{j-1}e_{j+2}e_{j+1}e_j + e_{j-1}e_{j-2}e_{j+1}e_j + e_{j-2}e_{j+1}e_j e_{j-1} - e_j e_{j-1}e_{j+1}e_{j+2} - e_{j-1}e_{j+1}e_j e_{j+2}\\
&- e_{j-1}e_j e_{j+2}e_{j+1} - e_{j-1}e_{j-2}e_{j+1}e_{j+1} - e_{j-2}e_j e_{j-1}e_{j+1} - e_{j-2}e_{j-1}e_{j+1}e_j + e_{j-1}e_j e_{j+1}e_{j+2}\\
&+ e_{j-2}e_{j-1}e_j e_{j+1} - d e_{j+1}e_j e_{j-1} + d e_{j-1}e_j e_{j+1} + e_{j+1}e_j + e_j e_{j-1} - e_j e_{j+1} - e_{j-1}e_j,\end{aligned} \tag{A.5}$$

$$\begin{aligned}J_{4,2}(j) = &-2e_{j+2}e_{j+1}e_j e_{j-1} + 2e_{j+1}e_j e_{j-1}e_{j+2} + 2e_j e_{j-1}e_{j+2}e_{j+1} + 2e_{j-1}e_{j+2}e_{j+1}e_j\\
&- 2e_j e_{j-1}e_{j+1}e_{j+2} - 2e_{j-1}e_{j+1}e_j e_{j+2} - 2e_{j-1}e_j e_{j+2}e_{j+1} + 2e_{j-1}e_j e_{j+1}e_{j+2}\\
&- d e_{j+1}e_j e_{j-1} - d e_j e_{j-1}e_{j+1} + d e_{j-1}e_{j+1}e_j + d e_{j-1}e_j e_{j+1} + 2e_{j+1}e_j - 2e_j e_{j+1},\end{aligned} \tag{A.6}$$

$$\begin{aligned}J_{4,3}(j) = &2(e_{j+3}e_{j+2}e_{j+1}e_j e_{j-1} + e_{j+2}e_{j+1}e_j e_{j-1}e_{j-2} - e_{j+2}e_{j+1}e_j e_{j-1}e_{j+3} - e_{j+1}e_j e_{j-1}e_{j+3}e_{j+2}\\
&- e_{j+1}e_j e_{j-1}e_{j-2}e_{j+2} - e_j e_{j-1}e_{j+3}e_{j+2}e_{j+1} - e_j e_{j-1}e_{j-2}e_{j+2}e_{j+1} - e_{j-1}e_{j+3}e_{j+2}e_{j+1}e_j\\
&- e_{j-1}e_{j-2}e_{j+2}e_{j+1}e_j - e_{j-2}e_{j+2}e_{j+1}e_j e_{j-1} + e_{j+1}e_j e_{j-1}e_{j+2}e_{j+3} + e_j e_{j-1}e_{j+2}e_{j+1}e_{j+3}\\
&+ e_j e_{j-1}e_{j+1}e_{j+3}e_{j+2} + e_j e_{j-1}e_{j-2}e_{j+1}e_{j+2} + e_{j-1}e_{j+2}e_{j+1}e_j e_{j+3} + e_{j-1}e_{j+1}e_j e_{j+3}e_{j+2}\\
&+ e_{j-1}e_j e_{j+3}e_{j+2}e_{j+1} + e_{j-1}e_{j-2}e_{j+1}e_j e_{j+2} + e_{j-1}e_{j-2}e_j e_{j+2}e_{j+1} + e_{j-2}e_{j+1}e_j e_{j-1}e_{j+2}\\
&+ e_{j-2}e_j e_{j-1}e_{j+2}e_{j+1} + e_{j-2}e_{j-1}e_j e_{j+2}e_{j+1}e_j - e_j e_{j-1}e_{j+1}e_{j+2}e_{j+3} - e_{j-1}e_{j+1}e_j e_{j+2}e_{j+3}\\
&- e_{j-1}e_j e_{j+2}e_{j+1}e_{j+3} - e_{j-1}e_j e_{j+1}e_{j+3}e_{j+2} - e_{j-1}e_{j-2}e_j e_{j+1}e_{j+2} - e_{j-2}e_j e_{j-1}e_{j+1}e_{j+2}\\
&- e_{j-2}e_{j-1}e_{j+1}e_j e_{j+2} - e_{j-2}e_{j-1}e_j e_{j+2}e_{j+1} + e_{j-1}e_j e_{j+1}e_{j+2}e_{j+3} + e_{j-2}e_{j-1}e_j e_{j+1}e_{j+2})\\
&+ d(+3e_{j+2}e_{j+1}e_j e_{j-1} + e_{j+1}e_j e_{j-1}e_{j-2} - e_{j+1}e_j e_{j-1}e_{j+2} - e_j e_{j-1}e_{j+2}e_{j+1}\\
&+ e_j e_{j-1}e_{j-2}e_{j+1} - e_{j-1}e_{j+2}e_{j+1}e_j - e_{j-1}e_{j-2}e_{j+1}e_j - e_{j-2}e_{j+1}e_j e_{j-1} - e_j e_{j-1}e_{j+1}e_{j+2}\\
&- e_{j-1}e_{j+1}e_j e_{j+2} - e_{j-1}e_j e_{j+2}e_{j+1} - e_{j-1}e_{j-2}e_{j+1}e_j - e_{j-2}e_j e_{j-1}e_{j+1} + e_{j-2}e_{j-1}e_{j+1}e_j\\
&+ 3e_{j-1}e_j e_{j+1}e_{j+2} + e_{j-2}e_{j-1}e_j e_{j+1} - 2e_j e_{j-1}e_{j+1}e_j + 3e_{j+1}e_j + e_j e_{j-1}\\
&+ 3e_j e_{j+1} + e_{j-1}e_j) + d^2(e_{j-1}e_j e_{j+1} + e_{j+1}e_j e_{j-1} + 2e_j) + 4(e_j + e_{j+1}).\end{aligned} \tag{A.7}$$

The higher-order generalized currents have a more complicated structure.

The explicit expressions for the local conserved quantities of the Temperley-Lieb Hamiltonian were obtained by Nienhuis and Huijgen [56]. However, the local conserved quantities obtained in [56] differ from those obtained by the boost operation; they are linearly dependent, but the choice of the linear combination is different.

## B   Numerical checks

We performed exact diagonalization to check our conjectures. Our procedure included the following steps:

- We numerically constructed the transfer matrix of the XXZ representation, local charges and generalized current operators which are relevant to our factorization formula (69)–(73) using (50). Our definitions and conventions are listed in Section B.1 Then, we exactly diagonalize the transfer matrix and store the eigenstates for the local charges.

- For each eigenstate, we numerically computed the current mean values to obtain $\psi_{\alpha,\beta}$ and then calculated the prediction of short-range correlation functions from the factorization formula (69)–(73), and compared them to the numerics from exact diagonalization.

- We also checked the current mean values are actually reproduced by the formula (60) using the corresponding Bethe roots. We utilized the famous TQ relation and calculated the $Q$ function from the eigenvalue of the transfer matrix, and then we obtained the corresponding Bethe roots. This strategy is described in detail in [77].

- For other representations, the Potts, the golden, and the trace representation, we construct the eigenstates simultaneously diagonalizing the local charges and additional symmetries such as momentum and check the factorization formula in the same way as the case of the XXZ rep. If there are common eigenstates with the XXZ representation with the same energies and the same eigenvalues of the higher local charges, we checked the current mean values also coincide.

It was observed that our factorization formula (69)–(73) holds for all the eigenstates for the finite size system in all representations we give in this work.

## B.1 Definition of the transfer matrix in the XXZ representation

The transfer matrix for the XXZ model with periodic boundary conditions and twist $\phi$ is defined as:

$$T(u) = \text{Tr}_0 \left( R_{0L}(u) \dots R_{01}(u) \right), \tag{B.1}$$

where each index $i = 1, \dots L$ stands for a spin-1/2, and 0 is an auxiliary spin-1/2 which is traced over in the definition of $T(u)$. The $R$ matrix acting on the $i^{\text{th}}$ spin and the auxiliary spin is defined as:

$$R_{0i}(u) = e^{i \frac{\phi}{2L} \sigma_0^z} \left( \frac{\cos \frac{\gamma}{2} \sin(u + \frac{\gamma}{2})}{\sin \gamma} + \frac{\sin \frac{\gamma}{2} \cos(u + \frac{\gamma}{2})}{\sin \gamma} \sigma_0^z \sigma_i^z + (\sigma_0^- \sigma_i^+ + \sigma_0^+ \sigma_i^-) \right). \tag{B.2}$$

## B.2 Numerical data

We list some concrete numerical data in Table 1–14. In the upper table on each page, i.e. in Table 1, 3, 5, 7, 9, 11, 13, we list the values of the correlation functions considered in the factorization formula (69)–(73) w.r.t. several eigenstates. The detailed setups are explained in the next paragraphs. In the upper table on each page, we denote $\langle e_1 e_2 e_3 \rangle' \equiv \langle e_1 e_2 e_3 + e_3 e_2 e_1 \rangle$. In the lower table on each page, i.e., in Table 2, 4, 6, 8, 10, 12, 14, we list the values of the current mean values used in the factorization formula (69)–(73) w.r.t. the same eigenstates as those referred to in the corresponding upper tables. The errors of the correlation functions listed in the upper tables and the predicted values from the factorization formula (69)–(73) calculated by the current mean values listed in the lower tables are at most of the order of $10^{-9}$.

We list the values of the correlation functions related to the factorization formula (69)–(73) for several eigenstates in the XXZ representation for $L = 8$ and the corresponding Bethe roots in Table 1, 3, 5, 7, and list the corresponding current mean values in Table 2, 4, 6, 8. Table 1 and 2 are the zero-twist case. Table 3 and 4 correspond to the $Q = 3$ Potts case. Table 5 and 6 correspond to the golden chain case. Table 7 and 8 correspond to the trace representation with $d = 3$.

We also list examples of numerical data for the other representations. We consider the 3-state Potts representation, the golden chain, and the trace representation for $L = 8$, and consider $Q = 3$ for the Potts representation and $d = 3$ for the trace representation. We show the values of the correlation functions and the current mean values of the 3-state Potts representation in Table 11 and 12, and for the golden chain representation in Table 9 and 10, and for trace representation with $d = 3$ in Table 13 and 14.

We note that for the trace representation, the Hamiltonian is non-Hermitian, and we have to use the left and right eigenvectors for the calculation of the current mean values and the correlation functions.

Table 1: List of the correlation functions in the first 8 eigenstates of the XXZ representation calculated by exact diagonalization for $d = 0.35$, $\phi = 0$, total magnetization sector of $M = 0$, and $L = 8$. $E$ is the eigenenergy and $k$ is the overall momentum quantum number. We list the eigenstates with $0 \leq k \leq 4$ to resolve degeneracies caused by inversion symmetry. We also list the corresponding Bethe roots. We denote $\langle e_1 e_2 e_3 \rangle' \equiv \langle e_1 e_2 e_3 + e_3 e_2 e_1 \rangle$. The numerical errors of the correlation functions compared to those computed from the factorization formula (69)–(73), using the current mean values in Table 2, are at most $10^{-13}$. The third eigenstate marked with a star includes singular rapidities $i\gamma/2$.

| | $E$ | $k$ | $\langle e_1 e_2 \rangle$ | $\langle e_1 e_3 \rangle$ | $\langle e_1 e_2 e_3 \rangle'$ | $\langle e_1 e_4 \rangle$ | $\lambda_1$ | $\lambda_2$ | $\lambda_3$ | $\lambda_4$ |
|---|---|---|---|---|---|---|---|---|---|---|
| 1 | 6.2368 | 0 | 0.8020 | 0.5782 | 1.6032 | 0.6098 | $-0.7220$ | $-0.1795$ | 0.1795 | 0.7220 |
| 2 | 4.7474 | 4 | 0.6817 | $-0.0914$ | 0.9435 | 0.8249 | 1.5708i | $-0.2551$ | 0 | 0.2551 |
| 3* | 4.6747 | 4 | 0.4845 | 0.4747 | 1.2679 | $-0.1142$ | $-0.6974$i | $-0.1881$ | 0.1881 | 0.6974i |
| 4 | 4.5583 | 1 | $0.5222 + 0.2063$i | 0.1942 | 0.9298 | 0.3235 | $-0.7688 + 1.5708$i | $-0.1538$ | 0.1915 | 0.7311 |
| 5 | 3.5526 | 3 | $0.5550 - 0.0321$i | $-0.0375$ | 0.6384 | $-0.0798$ | $-0.3035 + 1.5708$i | $-0.3514$ | $-0.0980$ | 0.7529 |
| 6 | 3.4703 | 2 | $0.4184 + 0.1691$i | 0.2523 | 0.6000 | 0.0446 | $-0.4989 + 1.5708$i | $-0.4435$ | 0.2022 | 0.7402 |
| 7 | 3.4083 | 3 | $0.2689 - 0.0114$i | 0.2701 | 0.4907 | 0.2383 | $-0.2845 - 0.6975$i | $-0.1751$ | $-0.2845 + 0.6975$i | 0.7442 |
| 8 | 3.3539 | 2 | $0.3032 + 0.2103$i | 0.0395 | 0.2004 | 0.1721 | $-0.4630 - 0.6977$i | 0.1901 | $-0.4630 + 0.6977$i | 0.7358 |

Table 2: List of the current mean values in the XXZ representation used in the factorization formula (69)–(73). The parameters and the setup are the same as Table 1. The numerical errors of the current mean value formula (60) against the above numerics are at most $10^{-10}$ except for the third eigenstate with singular Bethe roots.

| | $\psi_{0,0}$ | $\psi_{0,1}$ | $\psi_{0,2}$ | $\psi_{0,3}$ | $\psi_{0,4}$ | $\psi_{1,1}$ | $\psi_{1,2}$ | $\psi_{1,3}$ | $\psi_{1,4}$ | $\psi_{2,2}$ | $\psi_{2,3}$ | $\psi_{2,4}$ | $\psi_{3,3}$ |
|---|---|---|---|---|---|---|---|---|---|---|---|---|---|
| 1 | 0.7796 | 0 | 1.0815 | 0 | 10.4254 | $-1.2991$ | 0 | $-7.0213$ | 0 | 8.8692 | 0 | 85.4961 | $-153.1087$ |
| 2 | 0.5934 | 0 | 1.8072 | 0 | 13.7250 | $-0.5219$ | 0 | $-8.5822$ | 0 | 5.9757 | 0 | 83.4127 | $-141.1362$ |
| 3* | 0.5843 | 0 | 1.8194 | 0 | 14.7781 | $-0.4285$ | 0 | $-8.4151$ | 0 | 6.0217 | 0 | 78.4259 | $-143.1089$ |
| 4 | 0.5698 | 0.4126i | 1.6891 | 0.3878i | 19.2954 | $-0.4778$ | $-1.2087$i | $-6.4624$ | $-17.0919$i | 6.9130 | 0.0233i | 72.5437 | $-140.0925$ |
| 5 | 0.4441 | $-0.0641$i | 0.7184 | $-5.0049$i | 9.4013 | $-0.7232$ | $-1.2983$i | $-3.8280$ | $-3.8111$i | 5.5405 | $-11.0802$i | 90.1072 | $-56.6526$ |
| 6 | 0.4338 | 0.3381i | 0.3301 | 3.2589i | $-7.8149$ | $-0.8489$ | 0.1415i | $-5.4473$ | 12.3512i | 3.7061 | 15.5615i | 28.0494 | $-93.2620$ |
| 7 | 0.4260 | $-0.0228$i | 0.6272 | $-4.7661$i | 6.4299 | $-0.6326$ | $-1.5619$i | $-3.5612$ | $-11.3436$i | 4.5125 | $-14.1238$i | 65.2281 | $-63.3650$ |
| 8 | 0.4192 | 0.4205i | 0.5331 | 4.0813i | 2.5638 | $-0.6781$ | $-0.0447$i | $-4.2107$ | 2.4632i | 3.8831 | 17.0993i | 45.2646 | $-71.6897$ |

Table 3: List of the correlation functions in the 8 eigenstates of the XXZ representation calculated by exact diagonalization for $d = \sqrt{3}$, $z = -1$, total magnetization sector of $M = 0$, and $L = 8$. $E$ is the eigenenergy and $k$ is the overall momentum quantum number. We list the eigenstates for which the TQ equation can be solved, with $0 \leq k \leq 2$ to resolve the degeneracy, and if there are still degeneracies, only one of the degenerate states was listed. We also list the corresponding Bethe roots. The numerical errors of the correlation functions compared to those computed from the factorization formula (69)–(73), using the current mean values in Table 4, are at most $10^{-9}$.

| | $E$ | $k$ | $\langle e_1 e_2 \rangle$ | $\langle e_1 e_3 \rangle$ | $\langle e_1 e_2 e_3 \rangle'$ | $\langle e_1 e_4 \rangle$ | $\lambda_1$ | $\lambda_2$ | $\lambda_3$ | $\lambda_4$ |
|---|---|---|---|---|---|---|---|---|---|---|
| 1 | 10.0809 | 0 | 1.2053 | 1.7059 | 2.4500 | 1.5374 | $-0.1783$ | $-0.0222$ | 0.1171 | 0.4927 |
| 2 | 7.9421 | 1 | $0.8162 + 0.2823i$ | 0.8367 | 1.3574 | 0.9696 | $-0.5795$ | $-0.0068$ | 0.1306 | 0.5258 |
| 3 | 7.4797 | 0 | 0.8192 | 0.6285 | 1.6802 | 0.7205 | $-0.5658 - 0.3534i$ | $-0.5658 + 0.3534i$ | $-0.0603$ | 0.0657 |
| 4 | 7.0595 | 2 | 0.6589 | 0.8285 | 0.5820 | 0.5595 | $-0.5470$ | $-0.1381$ | 0.1381 | 0.5470 |
| 5 | 6.1962 | 2 | $0.5458 + 0.4208i$ | 0.3010 | 0.4200 | 0.5750 | $0.3458 - 0.2891i$ | 0.0564 | 0.2747 | $0.3458 + 0.2891i$ |
| 6 | 5.8054 | 0 | 0.4385 | 0.5467 | 1.0597 | 0.3386 | $-0.5792$ | $0.1972 - 0.2614i$ | $-0.0033$ | $0.1972 + 0.2614i$ |
| 7 | 4.5898 | 0 | 0.2697 | 0.2594 | $-0.4439$ | 0.0679 | $-0.4953 - 0.3368i$ | $-0.4953 + 0.3368i$ | $-0.2421$ | 0.2268 |
| 8 | 4.4528 | 1 | $0.4040 + 0.1477i$ | 0.1972 | 0.8463 | 0.1012 | $0.2167 - 0.5333i$ | 0.0369 | $0.2167 + 0.5333i$ | 0.2313 |

Table 4: List of the current mean values in the XXZ representation with the parameters being the same as Table 3. The numerical errors of the current mean value formula (60) against the above numerics are at most $10^{-8}$.

| | $\psi_{0,0}$ | $\psi_{0,1}$ | $\psi_{0,2}$ | $\psi_{0,3}$ | $\psi_{0,4}$ | $\psi_{1,1}$ | $\psi_{1,2}$ | $\psi_{1,3}$ | $\psi_{1,4}$ | $\psi_{2,2}$ | $\psi_{2,3}$ | $\psi_{2,4}$ | $\psi_{3,3}$ |
|---|---|---|---|---|---|---|---|---|---|---|---|---|---|
| 1 | 1.2601 | 0 | 3.3463 | 0 | 31.6763 | $-2.9346$ | 0 | $-39.9194$ | 0 | 34.6966 | 0 | 1172.5214 | $-931.3222$ |
| 2 | 0.9928 | 0.5646i | 3.3830 | 10.0458i | 95.2807 | $-1.7078$ | $-0.0231i$ | $-18.8898$ | $-137.1473i$ | 34.2995 | 0.1941i | 1198.6269 | $-528.0020$ |
| 3 | 0.9350 | 0 | 4.9451 | 0 | 98.2431 | $-0.8162$ | 0 | $-31.1809$ | 0 | 27.9266 | 0 | 614.1580 | $-1106.1864$ |
| 4 | 0.8824 | 0 | 0 | 0 | $-121.1016$ | $-2.9061$ | 0 | $-40.3672$ | 0 | 3.7137 | 0 | $-171.0383$ | $-910.7546$ |
| 5 | 0.7745 | 0.8415i | 2.0736 | 15.6585i | 35.9072 | $-1.4575$ | 0.2653i | $-16.7290$ | $-32.4849i$ | 21.4550 | 79.1153i | 712.4336 | $-429.6805$ |
| 6 | 0.7257 | 0 | 3.3595 | 0 | 159.4623 | $-0.5310$ | 0 | 2.6678 | 0 | 34.2559 | 0 | 1208.7936 | $-126.7140$ |
| 7 | 0.5737 | 0 | $-1.5182$ | 0 | $-65.9139$ | $-2.3736$ | 0 | $-8.1776$ | 0 | 6.4392 | 0 | 272.2224 | $-8.7733$ |
| 8 | 0.5566 | 0.2953i | 2.9808 | 9.1305i | 115.1619 | $-0.3225$ | 1.2303i | $-4.7323$ | 63.9288i | 23.5032 | 74.0467i | 836.7167 | $-226.6202$ |

Table 5: List of the correlation functions in 8 eigenstates of the XXZ representation calculated by exact diagonalization for $d = \frac{1+\sqrt{5}}{2}$, $z = e^{i2\gamma}$, total magnetization sector of $M = 0$, and $L = 8$. $E$ is the eigenenergy and $k$ is the overall momentum quantum number. We list the eigenstates for which the TQ equation can be solved, with $0 \leq k \leq 4$ to resolve the degeneracy, and if there are still degeneracies, only one of the degenerate states was listed. We also list the corresponding Bethe roots. The numerical errors of the correlation functions compared to those computed from the factorization formula (69)–(73), using the current mean values in Table 6, are at most $10^{-11}$.

| | $E$ | $k$ | $\langle e_1 e_2 \rangle$ | $\langle e_1 e_3 \rangle$ | $\langle e_1 e_2 e_3 \rangle'$ | $\langle e_1 e_4 \rangle$ | $\lambda_1$ | $\lambda_2$ | $\lambda_3$ | $\lambda_4$ |
|---|---|---|---|---|---|---|---|---|---|---|
| 1 | 10.0259 | 0 | 1.2264 | 1.7212 | 2.4912 | 1.5016 | $-0.2745$ | $-0.0580$ | 0.1055 | 0.4036 |
| 2 | 7.9546 | 4 | 0.8090 | 1.0181 | 1.8625 | 0.6663 | $0.3081 - 0.3184i$ | $-0.0967$ | 0.0665 | $0.3081 + 0.3184i$ |
| 3 | 6.2683 | 3 | $0.5180 - 0.0180i$ | 0.6133 | 0.7581 | 0.6905 | $0.0797 - 0.3142i$ | $-0.0868$ | $0.0797 + 0.3142i$ | 0.3956 |
| 4 | 6.2025 | 2 | $0.5312 + 0.3708i$ | 0.3187 | 0.3476 | 0.6375 | $-0.0397 - 0.3142i$ | $-0.0397 + 0.3142i$ | 0.0966 | 0.4025 |
| 5 | 4.6077 | 4 | 0.2387 | 0.2136 | $-0.4355$ | 0.0837 | $0.1632 - 0.3142i$ | $-0.3152$ | $0.1632 + 0.3142i$ | 0.3873 |
| 6 | 4.1221 | 0 | 0.3851 | 0.1706 | 1.0268 | 0.0217 | $0.4136 - 0.6247i$ | $-0.0010$ | $0.4136 + 0.6247i$ | 0.4584 |
| 7 | 3.4399 | 1 | $0.2320 - 0.2680i$ | 0.0412 | 0.0510 | 0.1185 | $0.4503 - 0.6307i$ | $-0.1456$ | $0.4503 + 0.6307i$ | 0.4963 |
| 8 | 2.4143 | 0 | $-0.1593$ | 0.1490 | $-0.0909$ | 0.0357 | $-0.1660 - 0.3142i$ | $0.4358 - 0.3290i$ | $-0.1660 + 0.3142i$ | $0.4358 + 0.3290i$ |

Table 6: List of the current mean values in the XXZ representation with the parameters being the same as Table 5. The numerical errors of the current mean value formula (60) against the above numerics are at most $10^{-8}$ except for the eigenstates indexed by 3 and 4. The current mean values of the eigenstates indexed by 3 and 4 calculated by (60) have larger numerical errors up to the order of $10^{-3}$ because the corresponding Bethe roots include (small value)$+i\gamma/2$ and cause loss of digits in the calculation of the Gaudin matrix.

| | $\psi_{0,0}$ | $\psi_{0,1}$ | $\psi_{0,2}$ | $\psi_{0,3}$ | $\psi_{0,4}$ | $\psi_{1,1}$ | $\psi_{1,2}$ | $\psi_{1,3}$ | $\psi_{1,4}$ | $\psi_{2,2}$ | $\psi_{2,3}$ | $\psi_{2,4}$ | $\psi_{3,3}$ |
|---|---|---|---|---|---|---|---|---|---|---|---|---|---|
| 1 | 1.2532 | 0 | 2.7575 | 0 | 40.1762 | $-3.1122$ | 0 | $-30.4413$ | 0 | 35.7292 | 0 | 699.9198 | $-1041.1975$ |
| 2 | 0.9943 | 0 | 4.4673 | 0 | 58.7948 | $-1.0641$ | 0 | $-35.8730$ | 0 | 24.2381 | 0 | 636.4054 | $-977.7322$ |
| 3 | 0.7835 | $-0.0360i$ | 1.6763 | $-14.6067i$ | 23.9194 | $-1.5671$ | $-4.8238i$ | $-15.9397$ | $-64.9254i$ | 17.9797 | $-77.5466i$ | 514.1511 | $-421.2121$ |
| 4 | 0.7753 | $0.7416i$ | 1.4832 | $12.7286i$ | 10.2488 | $-1.6686$ | $-0.1011i$ | $-18.3237$ | $9.9267i$ | 15.8433 | $90.9083i$ | 386.2803 | $-477.2798$ |
| 5 | 0.5760 | 0 | $-1.2677$ | 0 | $-24.5509$ | $-2.1720$ | 0 | 1.5124 | 0 | 11.0947 | 0 | 359.2093 | 133.1447 |
| 6 | 0.5153 | 0 | 3.4066 | 0 | 139.3342 | 0.0497 | 0 | 0.2926 | 0 | 24.9705 | 0 | 1033.3667 | $-13.3474$ |
| 7 | 0.4300 | $-0.5360i$ | 0.7507 | $-12.0616i$ | $-46.6899$ | $-0.8600$ | $-0.8968i$ | $-19.5390$ | $79.3132i$ | 1.3187 | $-20.1408i$ | $-95.3254$ | $-444.0795$ |
| 8 | 0.3018 | 0 | 0.3445 | 0 | $-7.5902$ | $-0.1736$ | 0 | $-4.2120$ | 0 | $-1.6588$ | 0 | 141.3088 | 93.4591 |

Table 7: List of the correlation functions in 8 eigenstates of the XXZ representation calculated by exact diagonalization for $d = 3$, $z = e^{i2\gamma}$, total magnetization sector of $M = 0$, and $L = 8$. $E$ is the eigenenergy and $k$ is the overall momentum quantum number. We list the eigenstates for which the TQ equation can be solved, with $0 \le k \le 4$ to resolve the degeneracy, and if there are still degeneracies, only one of the degenerate states was listed. We also list the corresponding Bethe roots. We denote $\langle e_1 e_2 e_3 \rangle' \equiv \langle e_1 e_2 e_3 + e_3 e_2 e_1 \rangle$. The numerical errors of the correlation functions compared to those computed from the factorization formula (69)–(73), using the current mean values in Table 8, are at most $10^{-12}$.

| | $E$ | $k$ | $\langle e_1 e_2 \rangle$ | $\langle e_1 e_3 \rangle$ | $\langle e_1 e_2 e_3 \rangle'$ | $\langle e_1 e_4 \rangle$ | $\lambda_1$ | $\lambda_2$ | $\lambda_3$ | $\lambda_4$ |
|---|---|---|---|---|---|---|---|---|---|---|
| 1 | 14.7275 | 0 | 1.6655 | 4.3998 | 3.6248 | 2.6996 | $-0.1177-0.4815i$ | $-0.1177+0.4815i$ | $-0.0474+0.1232i$ | $-0.0474-0.1232i$ |
| 2 | 13.2170 | 4 | 1.0245 | 4.1074 | 3.1863 | 1.1395 | $-1.5652-1.5708i$ | $-0.0084+0.2508i$ | $-0.0084-0.2508i$ | $-0.0070$ |
| 3 | 10.4142 | 3 | $0.6402-0.1402i$ | 1.6553 | 1.1036 | 2.5303 | $-0.9135-0.2344i$ | $-0.0550+0.5535i$ | $0.0313-0.0726i$ | $0.0968-0.2466i$ |
| 4 | 10.0000 | 2 | $0.7000+0.5000i$ | 1.0500 | 0.3000 | 2.1000 | $-0.7742-0.3414i$ | $-0.0764+0.5237i$ | $-0.0175+0.1471i$ | $0.1867-0.3294i$ |
| 5 | 8.9038 | 0 | 0.1595 | 1.8310 | 2.1352 | 0.6915 | $-0.5422+1.5708i$ | $-0.6884$ | 0.0112 | 0.2740 |
| 6 | 8.0000 | 4 | 0.2727 | 0.6364 | $-0.5455$ | 0.6364 | $-0.7277$ | $-0.0668-0.5371i$ | $-0.0668+0.5371i$ | 0.2349 |
| 7 | 7.5858 | 1 | $0.1098-0.3902i$ | 0.5947 | 0.3964 | 1.4697 | $-0.6223-1.5383i$ | $-0.6377+0.1239i$ | $-0.0117-0.2804i$ | $0.3247+0.1239i$ |
| 8 | 5.3687 | 0 | $-0.2000$ | 0.1442 | $-0.2600$ | 0.1089 | $-0.6759+0.4202i$ | $-0.6759-0.4202i$ | $0.2793+0.4224i$ | $0.2793-0.4224i$ |

Table 8: List of the current mean values in the XXZ representation with the parameters being the same as Table 7. The numerical errors of the current mean value formula (60) against the above numerics are at most $10^{-10}$.

| | $\psi_{0,0}$ | $\psi_{0,1}$ | $\psi_{0,2}$ | $\psi_{0,3}$ | $\psi_{0,4}$ | $\psi_{1,1}$ | $\psi_{1,2}$ | $\psi_{1,3}$ | $\psi_{1,4}$ | $\psi_{2,2}$ | $\psi_{2,3}$ | $\psi_{2,4}$ | $\psi_{3,3}$ |
|---|---|---|---|---|---|---|---|---|---|---|---|---|---|
| 1 | 1.8409 | 0 | 4.7764 | 0 | 146.8741 | $-6.2902$ | 0 | $-77.6288$ | 0 | 114.3119 | 0 | 1840.2901 | $-5749.4570$ |
| 2 | 1.6521 | 0 | 7.4127 | 0 | 76.7147 | $-2.6713$ | 0 | $-132.1650$ | 0 | 74.1274 | 0 | 4818.2829 | $-3336.7489$ |
| 3 | 1.3018 | $-0.2803i$ | 3.8410 | $-37.3575i$ | 145.4547 | $-2.6036$ | $-12.0533i$ | $-34.1685$ | $42.6595i$ | 60.8878 | $-225.5422i$ | 3698.3625 | $-1340.4055$ |
| 4 | 1.2500 | $1.0000i$ | 2.0000 | $26.0000i$ | $-50.0000$ | $-3.6000$ | $-1.2000i$ | $-68.4000$ | $147.6000i$ | 28.4000 | $397.2000i$ | 650.8000 | $-2559.6000$ |
| 5 | 1.1130 | 0 | 7.1515 | 0 | 417.1100 | 0.8714 | 0 | $-1.6636$ | 0 | 85.6498 | 0 | 4379.9362 | $-1014.5115$ |
| 6 | 1.0000 | 0 | $-2.0000$ | 0 | 14.0000 | $-3.8182$ | 0 | 26.7273 | 0 | 46.0000 | 0 | 2198.0000 | 1324.9091 |
| 7 | 0.9482 | $-0.7803i$ | 0.6590 | $-18.8575i$ | $-158.9547$ | $-1.8964$ | $1.4467i$ | $-60.3315$ | $566.1595i$ | 3.6122 | $57.9578i$ | $-711.8625$ | $-1976.0945$ |
| 8 | 0.6711 | 0 | 0.3220 | 0 | $-55.7341$ | $-0.5812$ | 0 | $-22.7075$ | 0 | $-14.4617$ | 0 | 545.2737 | 385.9685 |

Table 9: List of the correlation functions in 8 eigenstates of the golden chain representation calculated by exact diagonalization for $L = 8$. $E$ is the eigenenergy and $k$ is the overall momentum quantum number. We list the eigenstates with $0 \leq k \leq 4$. The numerical errors of the correlation functions compared to those computed from the factorization formula (69)–(73), using the current mean values in Table 10, are at most $10^{-13}$. The eigenstates indexed $1, 4$ correspond to the eigenstates of the XXZ chain indexed $1, 2$ in Table 5 respectively. The errors between the corresponding correlation functions are at most of the order $10^{-13}$. The eigenstates other than those indexed $1, 4$ also correspond to eigenstates in the XXZ representation with different parametrization. For example, the eigenstates indexed $2, 3$ correspond to the XXZ representation with the twist $z = e^{4i\gamma}$ and the magnetization $M = 0$.

|   | $E$ | $k$ | $\langle e_1 e_2 \rangle$ | $\langle e_1 e_3 \rangle$ | $\langle e_1 e_2 e_3 \rangle'$ | $\langle e_1 e_4 \rangle$ |
|---|---|---|---|---|---|---|
| 1 | 10.0259 | 0 | 1.2264 | 1.7212 | 2.4912 | 1.5016 |
| 2 | 9.8473 | 4 | 1.1926 | 1.6279 | 2.4183 | 1.4663 |
| 3 | 9.5601 | 0 | 1.1441 | 1.4711 | 2.3027 | 1.4244 |
| 4 | 7.9546 | 4 | 0.8090 | 1.0181 | 1.8625 | 0.6663 |
| 5 | 7.7141 | 3 | $0.8007 + 0.2901i$ | 0.7735 | 1.3479 | 0.9368 |
| 6 | 7.5275 | 1 | $0.7862 - 0.2483i$ | 0.7364 | 1.2794 | 0.8243 |
| 7 | 7.3934 | 0 | 0.7990 | 0.6750 | 1.6799 | 0.6794 |
| 8 | 6.7599 | 2 | $0.6397 + 0.0425i$ | 0.7590 | 0.5768 | 0.5047 |

Table 10: List of the current mean values in the golden chain representation with the parameters being the same as Table 9. The eigenstates indexed $1, 4$ correspond to the eigenstates of the XXZ chain indexed $1, 2$ in Table 6 respectively. The errors between the corresponding mean values are at most of the order $10^{-12}$.

|   | $\psi_{0,0}$ | $\psi_{0,1}$ | $\psi_{0,2}$ | $\psi_{0,3}$ | $\psi_{0,4}$ | $\psi_{1,1}$ | $\psi_{1,2}$ | $\psi_{1,3}$ | $\psi_{1,4}$ | $\psi_{2,2}$ | $\psi_{2,3}$ | $\psi_{2,4}$ | $\psi_{3,3}$ |
|---|---|---|---|---|---|---|---|---|---|---|---|---|---|
| 1 | 1.2532 | 0 | 2.7575 | 0 | 40.1762 | −3.1122 | 0 | −30.4413 | 0 | 35.7292 | 0 | 699.9198 | −1041.1975 |
| 2 | 1.2309 | 0 | 2.9948 | 0 | 32.2644 | −2.8941 | 0 | −34.0877 | 0 | 32.8961 | 0 | 921.9507 | −876.4761 |
| 3 | 1.1950 | 0 | 3.3370 | 0 | 25.1091 | −2.5726 | 0 | −38.1117 | 0 | 29.3764 | 0 | 1094.4553 | −746.0619 |
| 4 | 0.9943 | 0 | 4.4673 | 0 | 58.7948 | −1.0641 | 0 | −35.8730 | 0 | 24.2381 | 0 | 636.4054 | −977.7322 |
| 5 | 0.9643 | 0.5801i | 3.3212 | 7.8011i | 86.8173 | −1.5634 | −0.6982i | −18.2058 | −120.7357i | 31.1574 | −1.7555i | 966.7497 | −546.4995 |
| 6 | 0.9409 | −0.4965i | 3.0183 | −10.7810i | 83.9916 | −1.6448 | −0.6943i | −16.0361 | 117.2059i | 30.5524 | −3.5068i | 1088.4021 | −376.7953 |
| 7 | 0.9242 | 0 | 4.6091 | 0 | 80.9245 | −0.8367 | 0 | −30.1360 | 0 | 24.9617 | 0 | 537.6382 | −981.7392 |
| 8 | 0.8450 | 0.0850i | 0.0164 | 1.3714i | −107.6978 | −2.7168 | 0.0721i | −36.1226 | 23.1560i | 3.7302 | 19.0219i | −139.8274 | −787.5306 |

Table 11: List of the correlation functions in 8 eigenstates of the Potts representation calculated by exact diagonalization for $Q = 3$ and $L = 8$. $E$ is the eigenenergy and $k$ is the overall momentum quantum number, and $z_q$ is the $Z_Q$ quantum number. We list the eigenstates with $0 \leq k \leq 2$ and $z_q = 0, 1$. We note that the physical system size here is $L/2 = 4$. The numerical errors of the correlation functions compared to those computed from the factorization formula (69)–(73), using the current mean values in Table 12, are at most $10^{-11}$. The eigenstates indexed $2, 4, 5, 6$ correspond to the eigenstates of the XXZ chain indexed $1, 2, 3, 4$ in Table 3 respectively. The errors between the corresponding correlation functions are at most of the order $10^{-10}$. The eigenstates other than those indexed $2, 4, 5, 6$ also correspond to eigenstates in the XXZ representation with different parametrization. For example, the eigenstates indexed $1, 3$ correspond to the XXZ representation with the twist $z = e^{2i\gamma}$ and the magnetization $M = 0$.

|   | $E$ | $k$ | $z_q$ | $\langle e_1 e_2 \rangle$ | $\langle e_1 e_3 \rangle$ | $\langle e_1 e_2 e_3 \rangle'$ | $\langle e_1 e_4 \rangle$ |
|---|---|---|---|---|---|---|---|
| 1 | 10.4050 | 0 | 0 | 1.2666 | 1.8830 | 2.5827 | 1.5995 |
| 2 | 10.0809 | 0 | 1 | 1.2053 | 1.7059 | 2.4500 | 1.5374 |
| 3 | 8.3849 | 0 | 0 | 0.8296 | 1.2029 | 1.9706 | 0.7067 |
| 4 | 7.9421 | 1 | 1 | $0.8162 + 0.2823i$ | 0.8367 | 1.3574 | 0.9696 |
| 5 | 7.4797 | 0 | 1 | 0.8192 | 0.6285 | 1.6802 | 0.7205 |
| 6 | 7.0595 | 2 | 1 | 0.6589 | 0.8285 | 0.5820 | 0.5595 |
| 7 | 6.7970 | 1 | 0 | $0.7641 + 0.2602i$ | 0.3195 | 1.2012 | 0.6615 |
| 8 | 6.6104 | 1 | 0 | $0.5281 + 0.0281i$ | 0.6812 | 0.7866 | 0.8062 |

Table 12: List of the current mean values in the Potts representation with the parameters being the same as Table 11. The eigenstates indexed $2, 4, 5, 6$ correspond to the eigenstates of the XXZ chain indexed $1, 2, 3, 4$ in Table 4 respectively. The errors between the corresponding mean values are at most of the order $10^{-7}$.

|   | $\psi_{0,0}$ | $\psi_{0,1}$ | $\psi_{0,2}$ | $\psi_{0,3}$ | $\psi_{0,4}$ | $\psi_{1,1}$ | $\psi_{1,2}$ | $\psi_{1,3}$ | $\psi_{1,4}$ | $\psi_{2,2}$ | $\psi_{2,3}$ | $\psi_{2,4}$ | $\psi_{3,3}$ |
|---|---|---|---|---|---|---|---|---|---|---|---|---|---|
| 1 | 1.3006 | 0 | 2.9079 | 0 | 45.7228 | −3.3411 | 0 | −33.2258 | 0 | 39.9918 | 0 | 771.4239 | −1229.7023 |
| 2 | 1.2601 | 0 | 3.3463 | 0 | 31.6763 | −2.9346 | 0 | −39.9194 | 0 | 34.6966 | 0 | 1172.5214 | −931.3222 |
| 3 | 1.0481 | 0 | 4.7128 | 0 | 60.2540 | −1.1768 | 0 | −41.0506 | 0 | 26.7520 | 0 | 775.6237 | −1112.7968 |
| 4 | 0.9928 | 0.5646i | 3.3830 | 10.0458i | 95.2807 | −1.7078 | −0.0231i | −18.8898 | −137.1473i | 34.2995 | 0.1941i | 1198.6269 | −528.0020 |
| 5 | 0.9350 | 0 | 4.9451 | 0 | 98.2431 | −0.8162 | 0 | −31.1809 | 0 | 27.9266 | 0 | 614.1580 | −1106.1864 |
| 6 | 0.8824 | 0 | 0 | 0 | −121.1016 | −2.9061 | 0 | −40.3672 | 0 | 3.7137 | 0 | −171.0383 | −910.7546 |
| 7 | 0.8496 | 0.5205i | 4.1880 | 11.2664i | 98.8764 | −0.9287 | 0.4442i | −22.1164 | −114.5883i | 28.0232 | 2.6114i | 1096.6866 | −528.9631 |
| 8 | 0.8263 | 0.0562i | 1.8294 | 16.1408i | 30.2047 | −1.6526 | 5.3119i | −17.0107 | 63.9912i | 20.3713 | 85.8017i | 631.6663 | −468.8777 |

Table 13: List of the correlation functions in 8 eigenstates of the trace representation calculated by exact diagonalization for $d = 3$. We restrict the eigensubspace of the $U(1)$ charges: $N_a = 0$ ($a = 1, 2, 3$) where $N_a$ is defined in the Eq. (50) in [55]. $E$ is the eigenenergy and $k'$ is the overall momentum quantum number concerning the two-site shift. We list the eigenstates with $0 \leq k' \leq 2$. The numerical errors of the correlation functions compared to those computed from the factorization formula (69)–(73), using the current mean values in Table 14, are at most $10^{-13}$. The eigenstates indexed $1, 2, 7$ correspond to the eigenstates of the XXZ chain indexed $1, 2, 3$ in Table 7 respectively. The errors between the corresponding correlation functions are at most of the order $10^{-13}$.

|   | $E$ | $k'$ | $\langle e_1 e_2 \rangle$ | $\langle e_1 e_3 \rangle$ | $\langle e_1 e_2 e_3 \rangle'$ | $\langle e_1 e_4 \rangle$ |
|---|---|---|---|---|---|---|
| 1 | 14.7275 | 0 | 1.6655 | 4.3998 | 3.6248 | 2.6996 |
| 2 | 13.2170 | 0 | 1.0245 | 4.1074 | 3.1863 | 1.1395 |
| 3 | 13.0858 | 0 | 1.4722 | 2.6731 | 2.8585 | 3.0251 |
| 4 | 11.7967 | 1 | 1.0923 + 0.5002i | 2.0904 | 1.9444 | 2.4491 |
| 5 | 10.9631 | 1 | 1.0293 + 0.2706i | 1.9718 | 1.7179 | 1.3104 |
| 6 | 10.5358 | 0 | 1.0130 | 1.5571 | 2.3080 | 1.2967 |
| 7 | 10.4142 | 1 | 0.6402 + 0.1402i | 1.6553 | 1.1036 | 2.5303 |
| 8 | 10.3673 | 2 | 0.8388 + 0.2242i | 2.0426 | 0.8651 | 1.0503 |

Table 14: List of the current mean values in the trace representation with the parameters being the same as Table 13. The numerical errors of the current mean value formula (60) against the above numerics are at most $10^{-10}$.

|   | $\psi_{0,0}$ | $\psi_{0,1}$ | $\psi_{0,2}$ | $\psi_{0,3}$ | $\psi_{0,4}$ | $\psi_{1,1}$ | $\psi_{1,2}$ | $\psi_{1,3}$ | $\psi_{1,4}$ | $\psi_{2,2}$ | $\psi_{2,3}$ | $\psi_{2,4}$ | $\psi_{3,3}$ |
|---|---|---|---|---|---|---|---|---|---|---|---|---|---|
| 1 | 1.8409 | 0 | 4.7764 | 0 | 146.8741 | −6.2902 | 0 | −77.6288 | 0 | 114.3119 | 0 | 1840.2901 | −5749.4570 |
| 2 | 1.6521 | 0 | 7.4127 | 0 | 76.7147 | −2.6713 | 0 | −132.1650 | 0 | 74.1274 | 0 | 4818.2829 | −3336.7489 |
| 3 | 1.6357 | 0 | 7.5679 | 0 | 83.0324 | −3.9041 | 0 | −118.2845 | 0 | 73.0298 | 0 | 4623.7150 | −3583.7330 |
| 4 | 1.4746 | 1.0003i | 6.8665 | 11.5587i | 255.2844 | −2.7927 | −4.9296i | −61.9843 | −402.8328i | 91.8175 | −10.2651i | 3458.0998 | −3489.5801 |
| 5 | 1.3704 | 0.5411i | 4.8218 | 33.9784i | 212.4193 | −3.4178 | 5.5017i | −44.7668 | −283.1773i | 85.5345 | 73.1007i | 4790.6704 | −1252.5544 |
| 6 | 1.3170 | 0 | 8.7549 | 0 | 307.2956 | −1.2956 | 0 | −72.0023 | 0 | 77.4292 | 0 | 2530.5220 | −4001.5256 |
| 7 | 1.3018 | 0.2803i | 3.8410 | 37.3575i | 145.4547 | −2.6036 | 12.0533i | −34.1685 | −42.6595i | 60.8878 | 225.5422i | 3698.3625 | −1340.4055 |
| 8 | 1.2959 | 0.4484i | 0.3683 | 11.3196i | −267.3614 | −4.9239 | 0.7804i | −96.6653 | 257.4877i | 23.2054 | 210.6901i | −237.9226 | −3551.3820 |

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
