# Peer review of "On correlation functions in models related to the Temperley-Lieb algebra"

_SciPost Physics, doi:SciPost Phys. 16, 003 (2024)_

## Round 2 · Referee Report · Anonymous (Referee 6) · 2023-11-1

Report

The authors study mean values of local operators generated by the Temperley-Lieb (TL) algebra underlying several integrable lattice models. The Hamiltonian $H=Q_2$ of these models is given as the sum of TL generators $e_j$ in different representations. This allows to obtain the commuting charges $Q_{\alpha>2}$ within the boost operator formalism. From these charges generalized current operators $J_{\alpha\beta}$ can be constructed which describe the flow of the charge densities $q_\alpha$ time-evolved by $Q_\beta$.

The best studied among these models is probably the XXZ spin chain. For this model short range correlation functions have be computed using their factorization property: this allows to express them in terms of products of the Taylor coefficients of two (two-point) functions $\omega$ and $\omega'$, called the 'physical part' depending on the model and eigenstate, and an 'algebraic part', depending only on the TL parameter $d$.

Focusing on singlet states of the local charges in the TL algebra or, equivalently, operators invariant under the action of the quantum group $U_q(sl(2))$ in the case of the XXZ spin chain the authors formulate two conjectures for the physical and algebraic part of the mean values, respectively. These conjectures connect the correlation functions in general TL models with the known results for the XXZ spin chain:
- the physical part of the mean values of generalized current operators is given by the Taylor coefficients of a function $\psi(x,y)$ related to the two-point function $\omega$. As shown in the earlier works [35-37] $\psi$ can be expressed in terms of the Bethe roots parameterising the eigenstate of the XXZ chain subject appropriately chosen twisted boundary conditions with the same values of local charges and with the same TL parameter $d$.
- the algebraic part for the states covered by the conjectures depends only on the TL parameter $d$ (but not on the eigenstate or the TL representation underlying the specific model).

The authors have checked their conjecture numerically for the translationally invariant expectation values of operators generated from $e_1,\dots,e_4$.

The conjectures connect mean values in singlet states of certain local operators generated by TL generators for models based on different representations of the TL algebra. This allows to use the known results from the theory of factorized correlation functions for the XXZ spin chain to the related correlators in the Potts model, an RSOS model (giving the golden chain in the Hamiltonian limit) and the trace representation of the TL algebra.
This observation extends the use of the factorization for a class of correlation functions beyond the six-vertex model. Moreover, the particularly simple simple form of the mean values of generalized currents may be useful in extending this this approach to integrable models beyond the ones considered in this paper.

I recommend to accept this paper for publication in Scipost Physics, after the points listed below have been addressed:

1- I suppose that Eq. (67) holds for singlet eigenstates $|E\rangle$ only (and the second function $\omega'$ enters in other states). If this is so the authors should emphasize this point in Conjecture 2. If not this should be explained.

2- The mean value $\langle e_1 e_4\rangle$, Eq. (72), appears to diverge for the Ising case ($d=\sqrt{2}$). The authors should discuss whether there is there any physics in this apparent divergence. Maybe this divergence is cancelled by an identity satisfied by the physical part of the expression (as observed in Refs. [9,53]).

3- Can the authors say which parameterisation (i.e. twist) of the XXZ model with $d=2\cos\pi/5$ gives the eigenvalues #2 and 3 for the golden chain (Table 9)? Similarly for the XXZ model with $d=2\cos\pi/6$ and the eigenvalues #1 and 3 for the $Q=3$ Potts model (Table 11)?

4- Caption of Table 10: should read 'current mean values of the GOLDEN CHAIN representation' instead of Potts.

---

## Round 2 · Referee Report · Anonymous (Referee 5) · 2023-11-3

Strengths

see report

Weaknesses

see report

Report

The paper "On correlation functions in models related to the Temperley-Lieb
algebra" by K.Fukai, R.Kleinemühl, B.Pozsgay and E.Vernier raises a two conjectures on the expression for two kinds of diagonal correlation functions, associated with singlet states of commuting charges in integrable models obtained from representations of the Temperley-Lieb algebra. The first conjecture provides a closed and explicit expression for the diagonal matrix elements of so-called "generalized current operators" in terms of an auxiliary function $\psi$ and its derivatives. The second conjecture expresses so called transitionally invariant mean values in terms of linear combinations of the same $\psi$ function and its derivatives.

The paper is in overall written in a clear fashion. There is only one issue that I'd like the authors to clarify. The claim in the conjectures states that the $\psi $ function is representation independent in the sense that for a given Eigenstate E, one should deduce this function from the one of the XXZ chain associated to an Eigenstate having the same value of local charges in the XXZ representation with the same TL parameter d.

First, how do the authors know that for a given TL representation with parameter d can they always find en Eigenstate on the XXZ chain side
having the claimed properties?

Second, it seems to me that the \omega function, say in (52), is the only representation dependent part in the formula (eg it changes whether one is in finite volume, infinite one, finite temperature, chosen averaging Eigenstate, etc). I think that it would be clearer if the authors managed to present their conjecture in terms of some object/quantity/formula that they directly attach to the representation for which the formula is written.

Finally, I list a few minor remarks/suggestions for corrections

(16) what is n?

Ref[2] on page 9 should be replaced by the original resut
B.M.~McCoy and T.T.~Wu, "Hydrogen-Bonded Crystals and the Anisotropic Heisenberg Chain.", Il Nuovo Cimento B, 56, (1968), 311-315.

(56) should end with a dot

It was unclear to me during reading whether (54) and (60) correspond to the same function ( omega was not defined explicitly in the paragraph related to the XXZ chain).
I would suggest to clarify that point somewhere around (60).

Requested changes

see report

---

## Round 2 · Referee Report · Anonymous (Referee 4) · 2023-11-17

Strengths

1- Clear conjecture supported by careful numerics.
2- Paper clearly written.

Weaknesses

1- The results are not totally unexpected given the existing literature.

Report

In this paper, the authors study correlation functions in models build from representations of the Temperley-Lieb algebra, in particular the Potts model, the golden-RSOS chain, or the trace representation. As is well-known, the spectrum of such Hamiltonians is contained in the XXZ spectrum. However degeneracies differ, which might affect correlation functions.

On the XXZ side several important results have been accumulated over the years. In particular, multiple integral representations for the mean values of local operators were established, and nice factorization properties were observed for those. It is natural to ask whether correlations functions in Temperley-Lieb generated models relate to known results for the XXZ spin chain.

To this effect, the authors present two conjectures, one regarding the form of the mean value of generalized current operators, the second one stating the factorization property of translationally invariant mean values of products of Temperley-Lieb generators. Both conjectures are thoroughly checked for several representations and eigenstates.

The paper is interesting and clearly written, I think it deserves publication. My main criticism is that the authors should motivate more some of the choices they make in their study. For instance, the focus on singlet states, discussing why shift invariance is important for the Potts model conjecture to hold, etc. Similarly, they should at least discuss whether the approach used to show factorization in XXZ would work directly or not in the models they study. In this regard the third to last and second to last paragraphs in the conclusion feel a bit lazy.

Requested changes

Besides the above, I have a few of minor comments

a) Page 4, replace 'can not' by 'cannot'.
b) Page 15, replace 'twits' by 'twist'.
c) Page 12, after equation (46): 'and $\delta_a(x)$ is proved to be zero' reads awkwardly.
d) 'the' is missing in section 3.4.

---

## Round 3 · Referee Report · Anonymous (Referee 2) · 2023-11-30

Strengths

see previous report

Weaknesses

see previous report

Report

No more comment. I suggest to publish.

---

## Round 3 · Referee Report · Anonymous (Referee 3) · 2023-12-12

Report

I believe the authors have addressed my concerns, I recommend publication.

---

## Round 3 · Referee Report · Anonymous (Referee 1) · 2023-12-13

Report

The authors have addressed the issues raised in my report on the original submission. The paper should be published in its present form.

---

## Round 3 · Author Response

We are thankful to the referees for the comments. Please find our replies below.

---

## Round 3 · List of Changes

To Referee 1

  1. We made conjecture 2 more precise, mentioning the singlet properties. At the moment we don't know about omega' in other (non-singlet) states.
  2. As pointed out by the referee, the apparent divergence of some correlators as $d\to 2$ (and, in fact, simularly for $d\to 1$) is cancelled by a cancellation of the numerator, as was previously observed in Refs [9,53]. We have commented on this fact at the end of Section 5.1.
  3. We added the explanation of the corresponding twist and magnetization for the eigenstates of Golden and Potts in the caption of Tables 9 and 11, as the referee requested.
  4. We fixed the typo pointed out in the caption of Table 10.

To Referee 2

Major comments from Referee 2

Q: "First, how do the authors know that for a given TL representation with parameter d can they always find en Eigenstate on the XXZ chain side having the claimed properties?" A: We added an explanation on the equivalence of periodic TL rep and XXZ. We also changed the Hamiltonian to the periodic one explicitly.

Q: "Second, it seems to me that the \omega function, say in (52), is the only representation dependent part in the formula (eg it changes whether one is in finite volume, infinite one, finite temperature, chosen averaging Eigenstate, etc). I think that it would be clearer if the authors managed to present their conjecture in terms of some object/quantity/formula that they directly attach to the representation for which the formula is written." A: We added some clarification about the omega function in multiple places, explaining that it can take different values depending on the physical situation, but the factorization is always the same. We hope it is clear now.

Minor remarks/suggestions for corrections

Q: "(16) what is n?" A: We fixed the typo in (16), changing n -> Q.

Q: "on page 9 should be replaced by the original resut B.M.~McCoy and T.T.~Wu, "Hydrogen-Bonded Crystals and the Anisotropic Heisenberg Chain.", Il Nuovo Cimento B, 56, (1968), 311-315." A: We added the suggested reference.

Q: "(56) should end with a dot" A: Typo fixed.

Q: "It was unclear to me during reading whether (54) and (60) correspond to the same function ( omega was not defined explicitly in the paragraph related to the XXZ chain). I would suggest to clarify that point somewhere around (60)." A: This is the same function indeed.

To Referee 3

Requested small changes, which are now fixed: (a) can not -> cannot (b) twits -> twist (c) Kohei erased and $\delta_\alpha(x)$ is proved to be zero. (d) we added the missing "the" in the title of the subsection 3.4.

Other comments, suggestions:

Q: "Similarly, they should at least discuss whether the approach used to show factorization in XXZ would work directly or not in the models they study. In this regard the third to last and second to last paragraphs in the conclusion feel a bit lazy." A: We expanded the third to last paragraph in the Conclusion, mentioning a possible direction for a proof. Also, we extended the paragraph afterwards.

Q: "For instance, the focus on singlet states, discussing why shift invariance is important for the Potts model conjecture to hold, etc." A: We added the explanation on singlet eigenstates and non-singlet eigenstates at the end of Section 5.1, furthermore we added some explanation about the relevance of the shift invariance to the Conclusions.

---

## Editorial Decision

published